# HOPS recognizes each SNARE, assembling ternary *trans*-complexes for rapid fusion upon engagement with the 4th SNARE

**Hongki Song, Amy S Orr, Miriam Lee[†], Max E Harner[‡], William T Wickner***

Department of Biochemistry and Cell Biology, Geisel School of Medicine at Dartmouth, Hanover, United States

**\*For correspondence:**
William.T.Wickner@dartmouth.
edu

**Present address:** [†]School of Life Sciences and Cell Logistics Research Center, Gwangju Institute of Science and Technology, Gwangju, Republic of Korea; [‡]Biomedical Center Munich, Institute of Cardiovascular Physiology and Pathophysiology, Ludwig-Maximillians University Munich, Planegg-Martinsried, Germany

**Competing interests:** The authors declare that no competing interests exist.

**Abstract** Yeast vacuole fusion requires R-SNARE, Q-SNAREs, and HOPS. A HOPS SM-family subunit binds the R- and Qa-SNAREs. We now report that HOPS binds each of the four SNAREs. HOPS catalyzes fusion when the Q-SNAREs are not pre-assembled, ushering them into a functional complex. Co-incubation of HOPS, proteoliposomes bearing R-SNARE, and proteoliposomes with any two Q-SNAREs yields a rapid-fusion complex with 3 SNAREs in a *trans*-assembly. The missing Q-SNARE then induces sudden fusion. HOPS can 'template' SNARE complex assembly through SM recognition of R- and Qa-SNAREs. Though the Qa-SNARE is essential for spontaneous SNARE assembly, HOPS also assembles a rapid-fusion complex between R- and QbQc-SNARE proteoliposomes in the absence of Qa-SNARE, awaiting Qa for fusion. HOPS-dependent fusion is saturable at low concentrations of each Q-SNARE, showing binding site functionality. HOPS thus tethers membranes and recognizes each SNARE, assembling R+Qa or R+QbQc rapid fusion intermediates.

## Introduction

Membrane fusion at each organelle is orchestrated by conserved families of proteins and lipids with complex binding relationships (*Wickner and Rizo, 2017*). Tethering effectors bind Rab family GTPases to hold membranes in apposition (*Baker and Hughson, 2016*). SNARE (soluble N-ethylmaleimide-sensitive-factor attachment receptor) proteins are found on both fusion partners, either in cis-SNARE complexes if all are anchored to one membrane or *trans*-SNARE complexes if anchored to two apposed membranes. SNAREs have heptad-repeat SNARE domains with a central arginyl (R) or glutaminyl (Q) residue. SNAREs are grouped by sequence homology into four families, R, Qa, Qb, and Qc (*Fasshauer et al., 1998*). SNARE complexes have one member each of the R, Qa, Qb, and Qc families, with their α-helical SNARE domains wrapped together in a coiled coil (*Sutton et al., 1998*). This 4-SNARE bundle is stabilized by the interior disposition of apolar residues, with the exception of 1 arginyl and three glutaminyl residues in the center of the SNARE domain, termed the 0-layer. SNARE complex assembly can be promoted by Sec1/Munc18 (SM) family proteins (*Rizo and Südhof, 2012*), which have conserved surface grooves to bind the R- and Qa-SNARE domains (*Baker et al., 2015*). Disassembly of the post-fusion *cis*-SNARE complexes is catalyzed by the ATP-driven chaperone Sec18/NSF, stimulated by its co-chaperone Sec17/αSNAP (*Weber et al., 1998*; *White et al., 2018*). Sec17 and Sec18 also function earlier, stimulating the fusion of docked membranes (*Song et al., 2017*; *Zick et al., 2015*). Fusion also requires acidic lipids and phosphoinositides to promote the binding of peripheral membrane fusion proteins (*Cheever et al., 2001*; *Mima and Wickner, 2009*; *Orr et al., 2015*), and fatty acyl fluidity (*Zick and Wickner, 2016*) and nonbilayer-prone lipids (*Zick et al., 2014*) to enable the bilayer rearrangements which are essential for fusion.

We study membrane fusion mechanisms with the vacuole (lysosome) of *Saccharomyces cerevisiae*. Vacuoles undergo constant fission and fusion, regulated by growth medium osmolarity. Mutations which block fusion allow continued fission, resulting in a visibly altered <u>va</u>cuole <u>m</u>orphology which allowed selection of *vam* mutants in fusion (*Wada et al., 1992*). The *VAM* genes encode proteins which are unique to vacuole fusion: the Rab GTPase Ypt7, the 6 subunits of the HOPS (<u>ho</u>motypic fusion and vacuole <u>p</u>rotein <u>s</u>orting) tethering and SM complex (*Nakamura et al., 1997*; *Seals et al., 2000*; *Wurmser et al., 2000*), and the Qa, and Qc SNAREs of this organelle (hereafter referred to as Qa and Qc). The R-SNARE Nyv1 was found later (*Nichols et al., 1997*) and other vacuole fusion proteins such as the Qb SNARE Vti1, Sec17, and Sec18 are required in the exocytic pathway and were not identified in the *vam* screen since their loss is lethal.

Vacuole fusion has been extensively studied in vivo, in vitro with the purified organelle, and as reconstituted with proteoliposomes bearing all-purified components (*Mima et al., 2008*; *Zick and Wickner, 2016*). The 'priming' stage of vacuole fusion, which precedes organelle association, entails phosphoinositide synthesis (*Mayer et al., 2000*) and Sec17- and Sec18- dependent *cis*-SNARE complex disassembly (*Mayer et al., 1996*). Priming is a prerequisite for tethering (*Mayer and Wickner, 1997*), which is largely mediated by the affinities of two of the HOPS subunits (Vps39 and Vps41) for the Rab Ypt7 on each vacuole membrane (*Brett et al., 2008*). Vacuole fusion differs in this regard from synaptic fusion, where vesicle tethering at the active zone of the plasma membrane precedes synaptic priming, which assembles SNARE into a release-ready state (*Südhof, 2013*). Vacuoles also have a 'back-up' tethering system through the affinity of the PX domain of the Qc SNARE for PtdIns3P in trans (*Zick and Wickner, 2014*). The Vps33 SM-family subunit of HOPS can catalyze the productive association of the R SNARE domain with the Qa SNARE domain, initiating the formation of a 4-SNARE complex (*Baker et al., 2015*; *Jiao et al., 2018*). Fusion can be supported by HOPS and SNAREs alone, but is further accelerated by Sec17 and Sec18p without requiring ATP hydrolysis (*Song et al., 2017*).

These fusion proteins and lipids show interdependent co-enrichment on docked vacuoles at a ring-shaped microdomain surrounding the directly apposed bilayers (*Wang et al., 2002*; *Fratti et al., 2004*). The full panoply of affinities and functional interactions of these fusion components is only now emerging. SM proteins are known to bind to Qa SNAREs, and a conserved R-SNARE-binding site has been found on the Vps33 subunit of HOPS and on other SM proteins (*Baker et al., 2015*). HOPS binds the inherently water-soluble Qc SNARE by the affinity of the Vps16 and Vps18 HOPS subunits (*Krämer and Ungermann, 2011*) for the PX region of Qc that is N-terminal to its SNARE domain (*Stroupe et al., 2006*). Direct affinity of HOPS for Qb has not been reported. In chemically defined subreactions of fusion, proteoliposomes bearing Ypt7 and the R-SNARE underwent HOPS-dependent assembly of all the Q-SNAREs, including Qb, into a 4-SNARE complex (*Orr et al., 2017*), and Vps33 protein was shown by single-molecule force spectroscopy to catalyze 4-SNARE assembly (*Jiao et al., 2018*). Once a 4-SNARE complex has assembled, several Sec17/αSNAP molecules can bind along its length (*Zhao et al., 2015*). The N-terminal apolar loop of SNARE-bound Sec17 has direct affinity for the lipid bilayer (*Zick et al., 2015*), while the membrane-distal C-terminus binds Sec18/NSF (*Marz et al., 2003*; *Winter et al., 2009*). HOPS also has direct affinity for phosphoinositides such as PtdIns3P (*Stroupe et al., 2006*) and for acidic lipids (*Karunakaran and Wickner, 2013*).

The availability of pure and active fusion proteins and their reconstitution into model subreactions has allowed the detection of additional functional affinities among these components. We now report that HOPS is a tethering and SNARE-assembly machine that not only binds the R and Qa-SNAREs through its SM subunit, but also binds the Qb and Qc SNAREs. These affinities support the assembly of rapid-fusion intermediates between membranes bearing Ypt7 and the R-SNARE and other membranes bearing Ypt7 and subsets of the three Q-SNAREs. These intermediates can be based on either the R- and Qa-SNAREs, which are recognized by the Vps33 HOPS subunit (*Baker et al., 2015*; *Jiao et al., 2018*), or on the novel combination of R-, Qb-, and Qc-SNAREs in the absence of Qa-SNARE. The capacity of HOPS to form these 3-SNARE intermediates is supported by the finding that HOPS binds directly to each SNARE. HOPS and these 3 SNAREs are in stable, isolable complexes, although it is not known whether each SNARE is only bound to HOPS or whether the SNAREs have begun engaging each other through coiled coils assembly of their SNARE domains. Upon encountering the third Q-SNARE, each complex supports strikingly rapid fusion. Without HOPS, proteoliposomes with any two Q-SNAREs are extremely slow to assemble with the

third and there are no rapid-fusion intermediates. As a complementary demonstration of the functionality of HOPS recognition of each Q-SNARE, we show that the HOPS-mediated fusion of R- and single Q-SNARE proteoliposomes is saturable at low concentrations of the soluble forms of the other Q-SNAREs, whereas there is no saturation at these concentrations when HOPS is replaced by polyethylene glycol. Thus, HOPS recognition of each Q-SNARE supports its functional assembly with the others. It is unclear when each SNARE passes from HOPS association to coiled-coils SNARE:SNARE association.

## Results

In detergent micellar solution, SNAREs can spontaneously assemble into 4-SNARE complexes or subcomplexes (*Fukuda et al., 2000*). In the context of lipid bilayers, *trans*-SNARE complex assembly may be affected by the membrane anchoring of SNAREs, by membrane apposition through tethering, and by the affinity of HOPS for the SNAREs on each membrane. Tethering per se will support functional *trans*-SNARE formation between R- and Q-SNARE proteoliposomes if the 3 Q-SNAREs are preassembled (*Song and Wickner, 2019*); does tethering suffice if the Q-SNAREs are not preassembled?

### HOPS is required when any Q-SNARE is not preassembled

To study the functional intermediates in SNARE complex assembly, we assayed fusion without an added tether, with tethering by the physiological and multifunctional HOPS complex bound to the Rab Ypt7 on each membrane, or with a simple synthetic tether. Our synthetic tether consists of dimeric glutathione S-transferase (GST) fused to a PX domain that can bind to PtdIns3P in each proteoliposomal membrane (*Song and Wickner, 2019*). Proteoliposomes bearing Ypt7 and R-SNARE with lumenally entrapped biotinylated phycoerythrin were mixed with proteoliposomes bearing Ypt7 and the 3 Q-SNAREs with entrapped Cy5-streptavidin. These mixed proteoliposomes were incubated without tethering agent, with HOPS, or with GST-PX. Either tethering agent sufficed for fusion, which was detected by the FRET from the mixing of the lumenal dyes (*Figure 1A*). Similar proteoliposomes in which the inherently water-soluble Qc SNARE bore a synthetic C-terminal transmembrane (tm) anchor (*Xu and Wickner, 2012*) showed similar fusion (*Figure 1B*). Although tethering is required, this shows that productive association of the three pre-assembled Q-SNAREs and the R-SNARE into functional *trans*-SNARE complex does not require catalysis by the Vps33 SM-protein subunit of HOPS, consistent with earlier findings that deletion within the R and Qa SNARE recognition domains of the Vps33 subunit of HOPS still permits fusion between R- and QaQbQc-SNARE proteoliposomes (*Baker et al., 2015*). In contrast, GST-PX does not suffice when one fusion partner bears only two Q-SNAREs and the third Q-SNARE is added in soluble form, but HOPS supports such fusion (*Figure 1C–E*), suggesting that HOPS helps recruit each Q-SNARE.

### Limited spontaneous Qc-SNARE assembly

Since wild-type Qa and Qb are membrane anchored, and only Qc is soluble, we analyzed the fusion of R- and QaQb-proteoliposomes with soluble Qc in more depth. The full kinetic time course shows that the synthetic tether GST-PX did support detectable fusion of R- and QaQb- proteoliposomes with added Qc, but only very slowly and after a 10 min lag (*Figure 1E*, red curve, and *Figure 1—figure supplement 2*, curve a). This lag was eliminated and the fusion rate enhanced by a 30 min preincubation of the QaQb-SNARE proteoliposomes with Qc-SNARE prior to addition of the GST-PX tether (*Figure 1—figure supplement 2*, curves a vs. c and g), revealing a capacity for slow spontaneous assembly of stable 3Q-SNARE complex from QaQb-proteoliposomes and Qc (curve g). This was not seen with GST-PX and 2Q-SNARE proteoliposomes lacking Qa or Qb and supplemented with sQa or sQb, respectively (*Figure 1C and D*), suggesting that these assembly events are kinetically too slow and/or thermodynamically unfavorable.

As a second, complementary assay for spontaneous assembly of functional 3Q-SNARE complex, we employed the R-SNARE without its membrane anchor, termed soluble-R (sR), a known fusion inhibitor (*Thorngren et al., 2004*; *Zick and Wickner, 2014*; *Song and Wickner, 2019*). Inhibition by sR can employ two mechanisms: 1. sR may compete for the conserved R-SNARE binding groove on the Vps33 subunit of HOPS (*Baker et al., 2015*), and 2. If a stable SNARE complex is assembled which includes the sR-SNARE, subsequent fusion with R-SNARE proteoliposomes is blocked; for

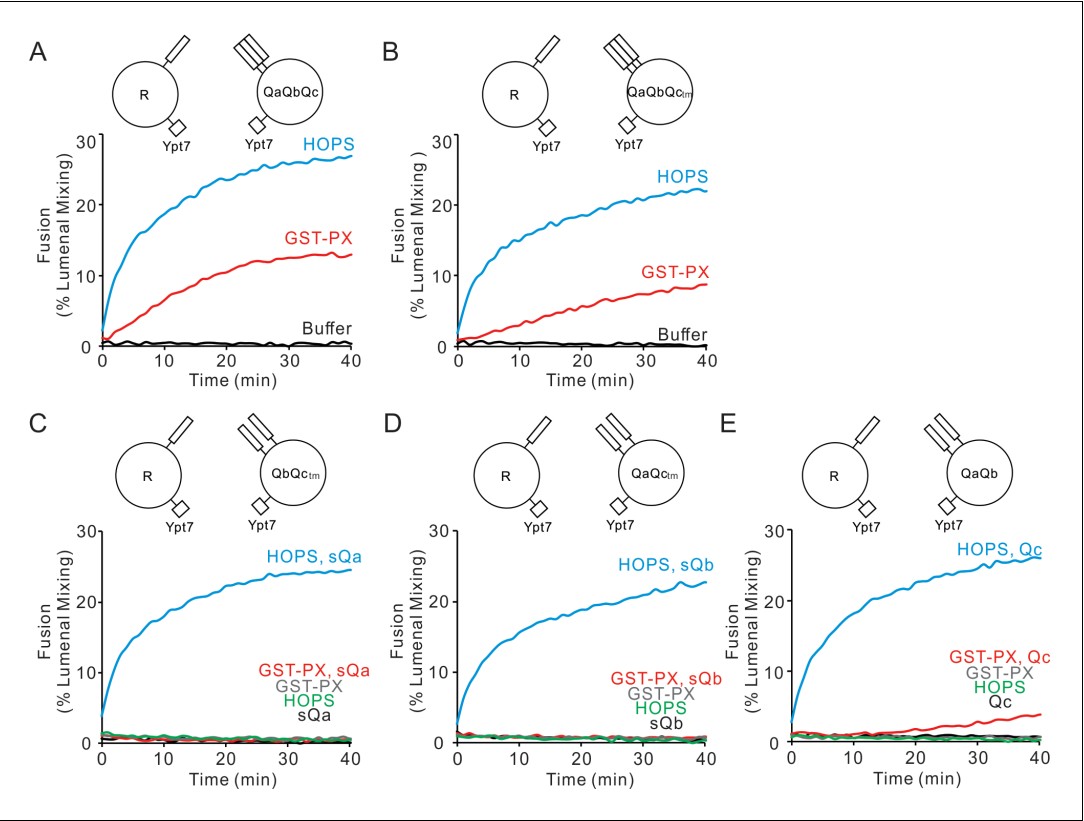

**Figure 1.** HOPS recruits each Q-SNARE, whereas a simple tether (GST-PX) does not. (A–E) Fusion reactions had proteoliposomes bearing either R- or Q-SNARE combinations as indicated at 1:16000 SNARE:lipid molar ratio. Fusion was assayed between R and (A) QaQbQc, (B) QaQbQc$_{tm}$, (C) QbQc$_{tm}$, (D) QaQc$_{tm}$, or (E) QaQb proteoliposomes as described in Materials and methods. Fusion reactions had 500 nM GST-PX or 50 nM HOPS as indicated. (A, B) Mixed proteoliposomes were incubated with HOPS (blue), GST-PX (red), or buffer (black). (C–E) HOPS or GST-PX and 4 μM soluble Q-SNAREs (sQ) were added: HOPS and sQ (blue), GST-PX and sQ (red), GST-PX alone (gray), HOPS alone (green), sQ alone (black). All proteoliposomes had Ypt7-tm at a 1:8000 protein:lipid molar ratio. Kinetic curves of content mixing assays in this figure are representative of n ≥ 3 experiments; average and standard deviations of fusion from three independent experiments are in *Figure 1—figure supplement 1*. The online version of this article includes the following source data and figure supplement(s) for figure 1:

**Source data 1.** Source data file (Excel) for *Figure 1A,B,C,D and E*.
**Figure supplement 1.** HOPS, but not GST-PX, supports fusion of R- and 2Q-SNARE proteoliposomes mixed with the third soluble Q-SNARE.
**Figure supplement 2.** The Qc-SNARE can slowly and stably assemble spontaneously with QaQb-proteoliposomes.
**Figure supplement 3.** Fusion inhibition by sR.

example, preincubation of Ypt7/3Q- and Ypt7/R-proteoliposomes with sR for 30 min prior to HOPS addition blocks their fusion (*Zick and Wickner, 2014*). R- and QbQc$_{tm}$ or QaQc$_{tm}$-SNARE proteoliposomes were mixed and preincubated with or without sR and with or without the third sQ for 30 min, then HOPS and (where absent) the sQ were added to initiate fusion (*Figure 1—figure supplement 3, A and B*). The fusion seen without sR (curves e, f) was inhibited approximately twofold by sR (curves a-d) without regard to the order of addition and incubation, which may reflect sR competition for a conserved site on Vps33. However, there was complete fusion inhibition when QaQb-proteoliposomes were preincubated with both sR and Qc for 30 min prior to HOPS addition (C, curve a), suggesting sRQaQbQc assembly. The contrast between the full inhibition by sR when preincubated with Qc and QaQb proteoliposomes (*Figure 1—figure supplement 3, C*, curve a) and the lack of inhibition enhancement when either soluble Qa or soluble Qb is preincubated with sR and QbQc$_{tm}$ or QaQc$_{tm}$ proteoliposomes, respectively (*Figure 1—figure supplement 3A and B*, curve

a), is another indication that soluble Qa and Qb do not spontaneously enter into complex with the other Q-SNAREs prior to HOPS addition. In sum, HOPS is required for any one of the Q-SNAREs to assemble rapidly with the others into functional SNARE complex for fusion, though a very slow spontaneous assembly of Qc can occur in the absence of HOPS.

## HOPS affinity for each SNARE

The above studies show that HOPS can support the integration of each Q-SNARE for fusion (*Figure 1C–E*), but do not address whether HOPS has the capacity to bind each SNARE directly. Prior studies have shown that HOPS has direct affinity for the PX domain of the Qc SNARE (*Stroupe et al., 2006*) through the HOPS Vps16 and Vps18 subunits (*Krämer and Ungermann, 2011*) and for the R- and Qa-SNARE domains through its Vps33 SM-family subunit (*Baker et al., 2015*). HOPS has not been reported to have direct affinity for the Qb SNARE. To evaluate the ability of HOPS to bind to each SNARE, we prepared six sets of liposomes, either protein-free liposomes or proteoliposomes bearing one of the four vacuolar SNAREs (including a characterized membrane-anchored form of Qc; *Xu and Wickner, 2012*) or all four SNAREs. Each set of proteoliposomes was incubated with HOPS, then mixed with density medium, overlaid with a density gradient, and subjected to ultracentrifugation. The floated proteolipsomes were assayed by immunoblot for bound HOPS. Although HOPS was not recovered with protein-free liposomes (*Figure 2*, lane 1; also *Figure 2—figure supplement 1*), HOPS bound to each of the 4 SNAREs (lanes 2–5) or their complex (lane 6). There are several possible reasons why HOPS may bind better to individual SNAREs than to the SNARE complex. Though the large apolar surfaces of the R- and Qa-SNARE domains can bind into grooves on the Vps33 surface (*Baker et al., 2015*), these same surfaces are oriented into the center of the 4-helical SNARE complex (*Sutton et al., 1998*). This may shield these apolar surfaces and thereby attenuate their contribution to HOPS binding the 4-SNARE complex. Furthermore, HOPS binds to Qc through the Qc N-domain (*Stroupe et al., 2006*). The interactions among the SNARE N-domains in a 4-SNARE complex may modulate the contribution of the Qc N-domain to binding the SNARE complex to HOPS.

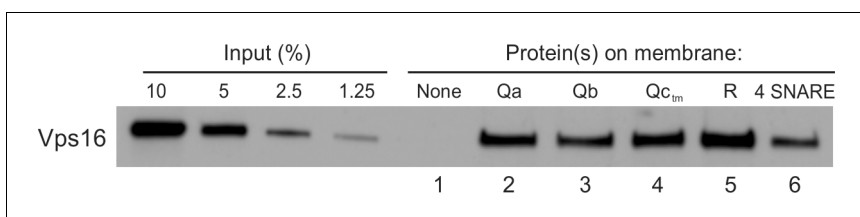

**Figure 2.** HOPS binds directly to each vacuolar SNARE. PC liposomes with no integral SNAREs, with each individual integrally-bound SNARE, or with all four wild-type SNAREs were incubated with HOPS at a twofold molar excess to SNAREs and floated. Flotation assays were conducted as described (*Orr et al., 2017*) with modifications. Liposomes (7.5 μl) were incubated for 1 hr at 30°C in 30 μl reactions (0.5 mM lipid, 500 nM HOPS, 0.2% defatted bovine serum albumin (BSA; Sigma-Aldrich), and 1 mM $MgCl_2$ in RB150). Reactions were gently vortexed with 90 μl of 54% (wt/vol) Histodenz (Sigma-Aldrich) in iso-osmolar RB150/$Mg^{2+}$ (containing a reduced level (2%) of glycerol) and 80 μl were transferred to 7 × 20 mm polycarbonate tubes (Beckman Coulter, Brea CA), overlaid with 80 μl of 35%, then 80 μl of 30% Histodenz in iso-osmolar RB150+$Mg^{2+}$ and finally 50 μl of RB150 +$Mg^{2+}$. The remaining portions of the starting incubations were solubilized with 1 μl of 5% (vol/vol) Thesit for determination of lipid recovery. Reactions were centrifuged in a Beckman TLS-55 rotor, 4°C, 55,000 rpm, 30 min. Samples were harvested by pipetting 80 μl from the top of the tube and solubilized with 2 μl of 5% Thesit. Lipid recovery was assayed as described (*Orr et al., 2015*), measuring either rhodamine fluorescence (excitation, 560 nm; emission, 580 nm; cutoff 570) or NBD fluorescence (excitation, 460 nm; emission, 538 nm; cutoff 515), depending on the composition of the liposomes. Bound protein determination was performed as described (*Orr et al., 2015*) by immunoblot of its Vps16 subunit with a standard curve of the input. Quantification and statistical analysis of HOPS binding from three independent experiments is in *Figure 2—figure supplement 1*. The online version of this article includes the following figure supplement(s) for figure 2:

**Figure supplement 1.** HOPS binds to each SNARE.

## HOPS assembles R- and Qa-SNARE fusion intermediates

Since Qc is the one physiologically soluble Q-SNARE, we sought to physically and functionally measure any HOPS stabilized fusion-competent assemblies in trans when Ypt7/R-SNARE proteoliposomes were incubated with Ypt7/QaQb proteoliposomes in the presence or absence of HOPS. Fusion required both HOPS and Qc (*Figure 3A*, curve d; other incubations initially lacked HOPS or Qc or both). After 30 min of incubation of mixed proteoliposomes with HOPS alone, the addition of Qc (red curve e) triggered very rapid fusion which was not seen without Qc (curve c), showing that a highly active fusion intermediate had accumulated. Samples from each incubation were withdrawn at 33 min, solubilized in detergent with an excess of GST-R to competitively block any wild-type R which might have otherwise associated with Qa in the extract, and assayed by pulldown with antibody to Qa for the amount of untagged R-SNARE which had become associated with the Qa-SNARE (*Figure 3B and C*). While there was background association of the R and Qa SNAREs in incubations without HOPS or Qc (lane a) and maximal association with both HOPS and Qc (lane d), HOPS promoted substantial *trans* complex assembly between R and Qa SNAREs in the absence of Qc (lane c) while fusion remained blocked (*Figure 3A*, curve c). The addition of Qc at 30 min triggered rapid

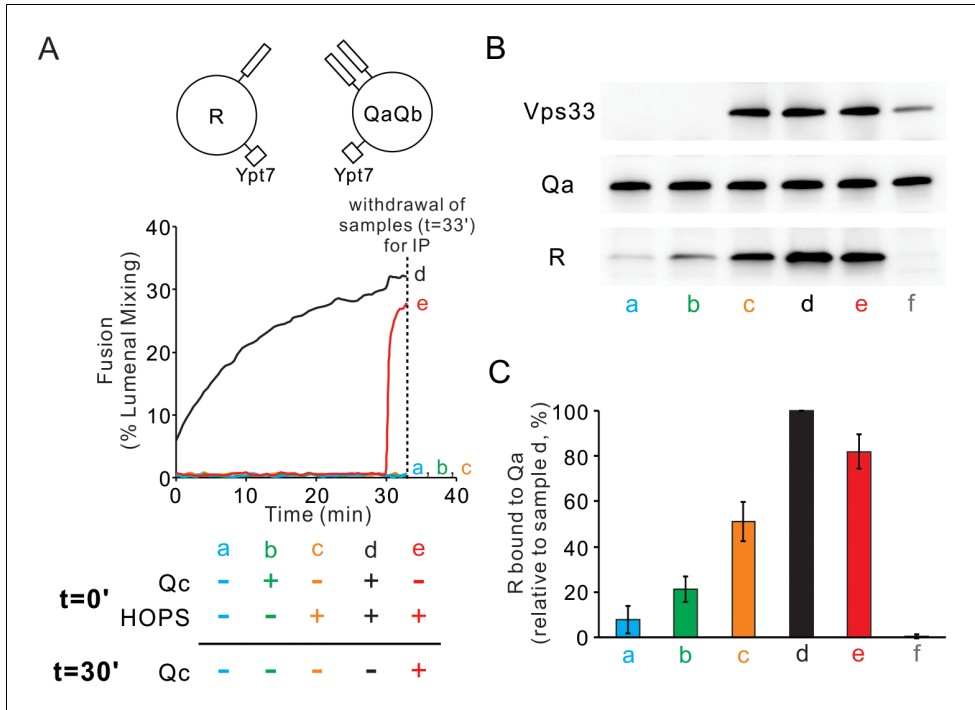

**Figure 3.** HOPS induces formation of a rapid-fusion intermediate which includes the R- and Qa-SNAREs in trans association with each other and/or the same HOPS molecule. Proteoliposomes with Ypt7 and R (1:8000 and 1:16,000 molar ratio to lipids, respectively) were mixed with proteoliposomes with Ypt7 and Qa and Qb SNAREs and with 50 nM HOPS and 4 µM Qc where indicated, added either at the start of incubation or after 30 min. (A) Fusion was assayed as lumenal FRET. After 30 min, Qc was added to one sample (e, red). (B) To measure complex formation, the amount of R SNARE that was immunoprecipitated from a detergent extract with anti-Qa antibody was determined. After incubation for 33 min, samples were placed on ice and mixed with five volumes of ice-chilled modified RIPA buffer [20 mM Hepes·NaOH, pH 7.4, 150 mM NaCl, 0.2% (wt/vol) BSA, 1% (vol/vol) Triton X-100,1% (wt/vol) sodium cholate, 0.1% (wt/vol) SDS] containing RIPA buffer-washed protein A magnetic beads (ThermoFisher), 5 µM GST-R and 5 µg anti-Qa antibody. After the mix was nutated at 4˚C for 2 hr, beads were washed three times with 1 mL of RIPA buffer. Proteins were eluted with 100 µL of SDS sample buffer at 95˚C for 5 min. Eluates were assayed by immunoblot with antibodies to R, Qa and Vps33. For sample f, the separate proteoliposomes, Qc, and HOPS were each mixed with RIPA buffer, then combined. (C) Immunoblots for the R-SNARE were scanned from five experiments, the band intensity of sample d (HOPS and Qc added at t = 0 min) was set to 100%, and the means and standard deviations are shown.

The online version of this article includes the following source data for figure 3:

**Source data 1.** Source data file (Excel) for *Figure 3A and C*.

fusion (*Figure 3A*, red curve e) with only a modest increase in trans complex (*Figure 3C,e* vs c). HOPS thus forms an assembly which includes the R- and Qa-SNAREs in trans, whether directly with each other in coiled coils 3-SNARE bundles or with the R- and the two Q- SNAREs associated with common HOPS molecules or by some combination of these associations. We refer to these rapid-fusion complexes as 'trans-complexes', since they include two proteins anchored to different membranes, and reserve the term 'trans-SNARE complex' for when the anchored SNAREs themselves are clearly in a coiled coils complex with each other.

Fusion mediated by the four SNAREs alone is blocked by Sec17, Sec18, and ATP, but these chaperones stimulate fusion in the presence of HOPS (*Mima et al., 2008*). Is the rapid-fusion intermediate which HOPS forms with Ypt7/R and Ypt7/QaQb proteoliposomes compatible with Sec17, Sec18, and ATP? Ypt7/R- and Ypt7/QaQb-SNARE proteoliposomes fuse when incubated with HOPS and Qc (*Figure 4A*, solid black curve a; also *Figure 4—figure supplement 1*). When Qc is withheld, there is no fusion, but upon its addition there is rapid fusion (dotted black curve d). Sec17/Sec18/ATP does not inhibit fusion, whether added from the start of incubations with HOPS and Qc (solid blue curve b) or after 25 min (solid red curve c). When Qc is withheld and only added after 30 min of incubation with HOPS (dotted black curve d), fusion is not diminished by the presence of Sec17/Sec18/ATP from the start of the incubation (dotted blue curve e) or when added after 25 min (dotted red curve f). This HOPS-dependent fusion intermediate is thus fully compatible with the Sec17/Sec18 SNARE disassembly chaperones.

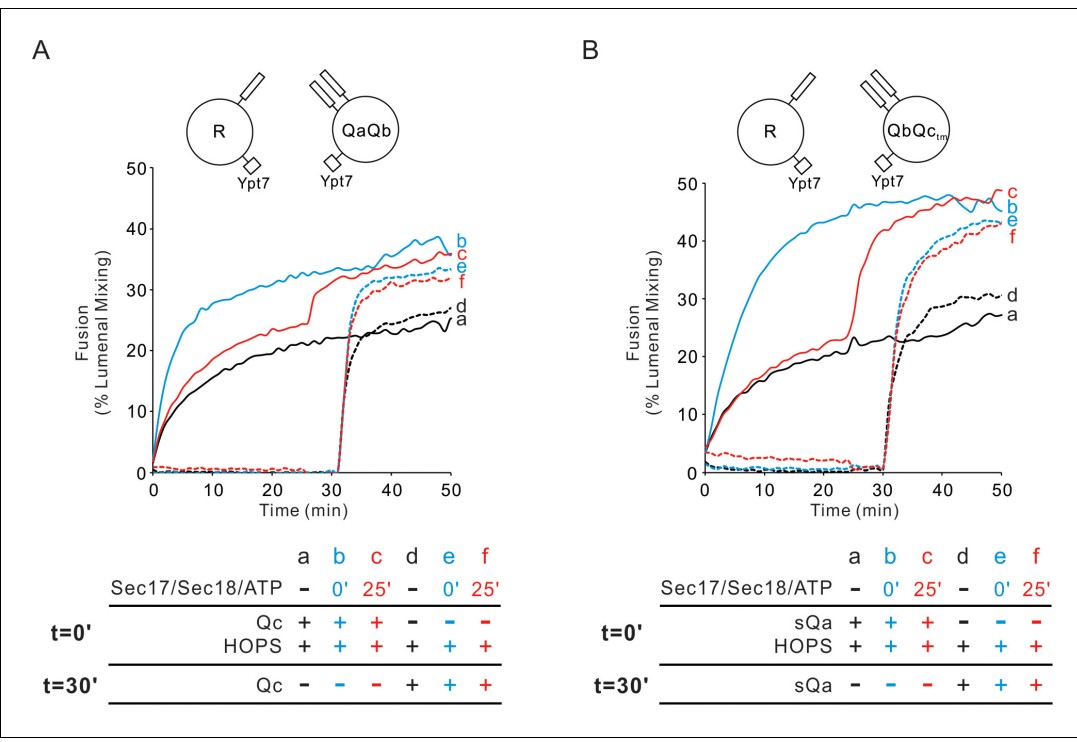

**Figure 4.** Sec17/Sec18/ATP do not inhibit fusion from the HOPS:R:QaQb or HOPS:R:QbQc$_{tm}$ intermediates. (**A**) Proteoliposomes bearing Ypt7 (1:8000 molar ratio to lipids) and either R- or QaQb- SNAREs (1:16,000 molar ratio to lipids) were mixed with 50 nM HOPS at t = 0, and 100 nM Qc was either added at t = 0 (**a–c**) or at t = 30 (**d–f**). Sec17 (300 nM), Sec18 (300 nM) and 1 mM ATP were added at t = 0 (**b, e**) or at t = 25 (**c, f**). Kinetics shown are representative of n ≥ 3 experiments. The average and standard deviations of maximum fusion rates from three independent experiments are in *Figure 4—figure supplement 1*. (**B**) The analogous experiment was performed with Ypt7/R and Ypt7/QbQc$_{tm}$ proteoliposomes with HOPS, sQa, and Sec17/Sec18/ATP as indicated.

The online version of this article includes the following source data and figure supplement(s) for figure 4:

**Source data 1.** Source data file (Excel) for *Figure 4A and B*.

**Figure supplement 1.** Rapid-fusion intermediates in the presence of Sec17, Sec18 and ATP.

To determine whether Qb was necessary for HOPS-dependent formation of this rapid-fusion intermediate between Ypt7/R and Ypt7/QaQb proteoliposomes, we prepared Ypt7/Qa proteoliposomes. Ypt7/Qa proteoliposomes can fuse with Ypt7/R proteoliposomes when provided HOPS, sQb without the Qb membrane anchor, and Qc (*Figure 5A*, curve a, and *Figure 5—figure supplement 1*), as reported (*Song et al., 2017*). When sQb, Qc, or both were omitted, fusion was blocked, but there was rapid fusion when the omitted Q-SNAREs were restored after the proteoliposomes had incubated for 30 min with HOPS (curves b-d). Rapid fusion required HOPS during the initial incubation period (curves e-h). Thus, the R- and Qa-SNAREs alone will suffice for a HOPS-dependent assembly of a rapid-fusion intermediate.

To test whether Qc SNARE will, when present, actually join in this HOPS:R:Qa rapid-fusion intermediate, we exploited Qc-3Δ, a mutant Qc SNARE which lacks its three C-terminal heptad repeats and thereby blocks fusion (*Schwartz and Merz, 2009*). The fusion of Ypt7/R and Ypt7/Qa proteoliposomes with sQb, Qc, and HOPS (*Figure 5B*, curve a) was blocked by a large molar excess of Qc-3Δ (curve b). Other incubations were performed with full-length Qc and HOPS but without Qb (curves c-i). Without Qc-3Δ, the addition of Qb after 30 min triggered rapid fusion (curve c). There was substantial resistance to inhibition by Qc-3Δ when it was added immediately before sQb (curve

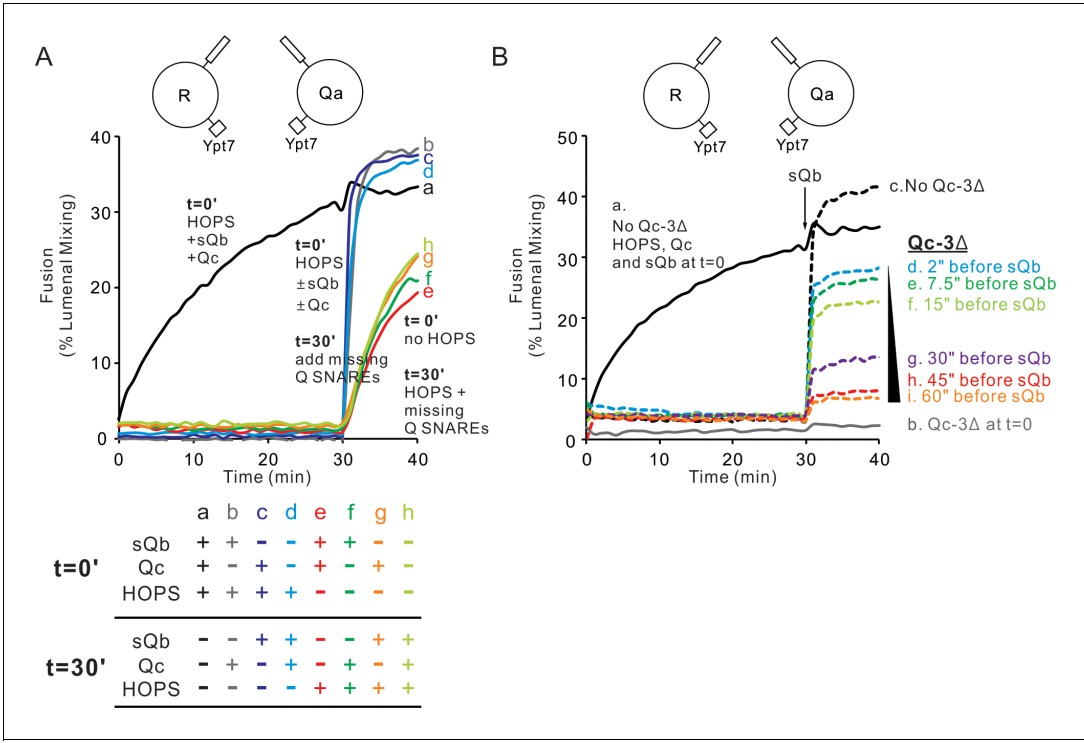

**Figure 5.** HOPS supports the assembly of a sudden-fusion intermediate between Ypt7/R-SNARE and Ypt7/Qa-SNARE proteoliposomes in the absence of Qb or Qc. (**A**) Mixed proteoliposomes bearing Ypt7 (1:8000 molar ratio to lipids) and either R- or Qa-SNARE (1:16,000 molar ratio to lipids) were mixed with 50 nM HOPS at t = 0 (**a–d**) or t = 30 min (**e–h**). Soluble Q SNAREs were added: (**a,e**) sQb and Qc at t = 0, (**b,f**) sQb at t = 0, and (**c,g**) Qc at t = 0. At t = 30′, all missing soluble SNAREs were added. (**B**) When Qc is present, it engages reversibly with the HOPS:R:Qa sudden-fusion complex. Proteoliposomes bearing Ypt7 (1:8000 molar ratio to lipids) and either R- or Qa- SNARE (1:16,000 molar ratio to lipids) were mixed and given 100 nM Qc, 4 μM Qb, 50 nM HOPS and/or 4 μM QcΔ3, then assayed for fusion as follows: (**a**) Qc, Qb and HOPS were added at t = 0, (**b**) Qc, Qb, HOPS and QcΔ3 (*Schwartz and Merz, 2009*) were added at t = 0, (**c–i**) Qc and HOPS were added at t = 0 and Qb was added at t = 30 min. For d-i, QcΔ3 was added 60 s (**d**), 45 s (**e**), 30 s (**f**), 15 s (**g**), 7.5 s (**h**) or 2 s (**i**) before Qb addition. Kinetics shown are representative of n ≥ 3 experiments. Average and standard deviations of maximum fusion rates from three independent experiments are in *Figure 5—figure supplement 1*.

The online version of this article includes the following source data and figure supplement(s) for figure 5:

**Source data 1.** Source data file (Excel) for *Figure 5A and B*.

**Figure supplement 1.** A sudden-fusion intermediate with R and Qa proteoliposomes.

d), but just 1 min of incubation with Qc-3Δ prior to sQb addition allowed full fusion inhibition (curve i). Had there been no association between wild-type Qc and the HOPS:R:Qa rapid-fusion machinery, the same fusion would have been seen in d-i, which each had the same 40-fold excess of Qc-3Δ to Qc at the time of sQb addition. The data indicate a labile association of wild-type Qc with the HOPS:R:Qa rapid-fusion machinery, taking a minute for full dissociation of Qc and capture by Qc-3Δ.

## Rapid-fusion intermediates for each sQ

To explore the role of HOPS in the assembly of each of the 3 Q-SNAREs into rapid-fusion intermediates, Ypt7/R-SNARE proteoliposomes were mixed with each of the three Ypt7/2Q-SNARE proteoliposomes, the soluble Q-SNARE that was not proteoliposome-bound, and HOPS (*Figure 6A–C*, black curve a). In each case, the rate of fusion was compared to that seen when HOPS had been pre-incubated with the two mixed sets of proteoliposomes without the soluble Q-SNARE for 30 min prior to soluble Q-SNARE addition (red curve b). For each Q-SNARE, the fusion was distinctly more rapid when the soluble Q-SNARE was added after 30 min (*Figure 6A–C*, red curve b; see *Figure 6—figure supplement 1*) than when it had been added from the start (curve a), indicating that HOPS allowed the accumulation of fusion-competent intermediates which included either R and QaQb, R and QaQc_{tm}, or R and QbQc_{tm}, respectively. The formation of the rapid-fusion state required the presence of HOPS during the preincubation (compare curve b to curves c, d). The action of HOPS is not merely due to its tethering function, as the synthetic tether GST-PX does not support fusion at all unless the three Q-SNAREs have been pre-assembled (*Figure 1*). These assays do not distinguish whether the R- and two Q-SNAREs had entered three-helical coiled coils SNARE subcomplexes or whether HOPS catalysis consisted of binding the R and two Q-SNAREs in a configuration which could rapidly and functionally receive the third Q-SNARE for fusion.

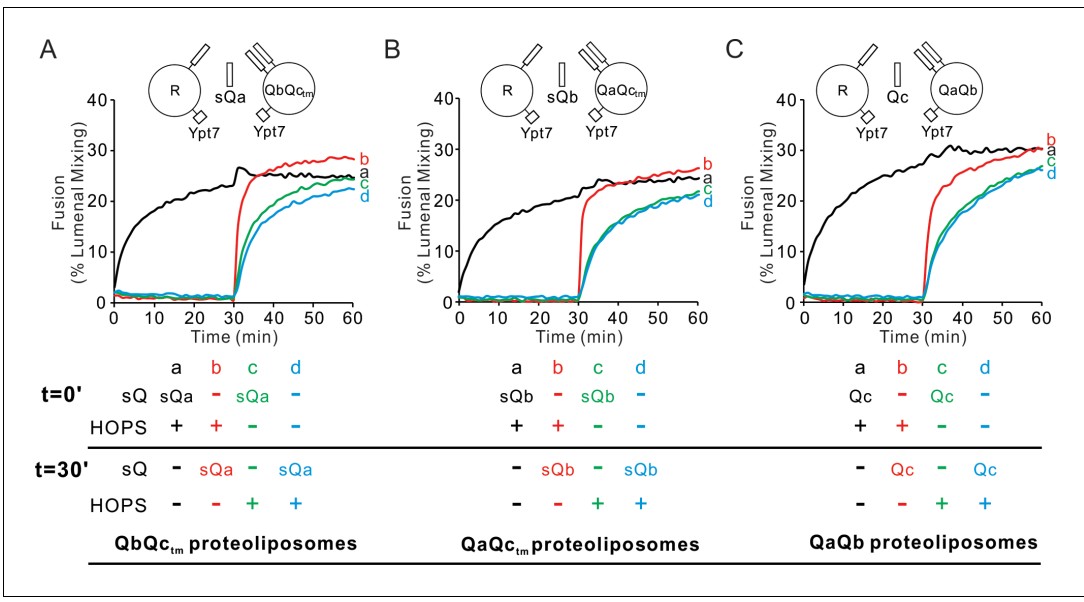

**Figure 6.** HOPS activates mixed proteoliposomes with R+Ypt7 and 2Q+Ypt7 for a burst of fusion when the missing sQ-SNARE is supplied. (A–C) Fusion incubations received 50 nM HOPS at t = 0 min (**a, b**) or t = 30 min (**c, d**) with 4 µM soluble Q-SNARE at t = 0 min (**a, c**) or t = 30 min (**b, d**) as indicated. Fusion reactions had proteoliposomes bearing R mixed with proteoliposomes bearing either (**A**) QbQc_{tm}, (**B**) QaQc_{tm}, or (**C**) QaQb SNAREs at 1:16000 SNARE:lipid molar ratios. All proteoliposomes had Ypt7-tm at a 1:8000 protein: lipid molar ratio. Content mixing assays in this figure are representative of n ≥ 3 experiments; means and standard deviations for each experiment are presented in *Figure 6—figure supplement 1*.

The online version of this article includes the following source data and figure supplement(s) for figure 6:

**Source data 1.** Source data file (Excel) for *Figure 6A,B and C*.

**Figure supplement 1.** Preincubation of R- and 2Q-SNARE proteoliposomes with HOPS gives more rapid fusion when the third soluble SNARE is added than when all components are mixed without preincubation.

Does each rapid-fusion complex correspond to a triad of SNAREs which alone can associate stably without HOPS? As reported (*Fukuda et al., 2000*), purified vacuolar SNAREs will associate in mixed micellar solution and can be isolated as a complex on affinity beads (*Figure 7A–C*, lanes 2). The single omissions of Qb, Qc, or R still allows formation of RQaQc, RQaQb, or QaQbQc complexes (*Figure 7A*, lanes 4–6). However, RQbQc complex is not seen when Qa is omitted (*Figure 7B and C*, lanes 2 vs 3). This need for Qa for spontaneous SNARE complex assembly, and the conserved recognition of Qa by SM proteins (*Baker et al., 2015*), makes the absence of Qa from the Ypt7/R and Ypt7/QbQc rapid-fusion intermediate of particular interest.

Since Ypt7/R and Ypt7/QbQc proteoliposomes which are incubated with HOPS will undergo rapid fusion when sQa is added (*Figure 6A*, and *Figure 8A*, red curve e), we assayed whether there was HOPS-dependent physical association between Qb- and R-SNARE prior to sQa addition. This association was indeed seen in a HOPS-dependent manner (*Figure 8B and C*, lanes a vs c), even though the R, Qb, and Qc SNAREs in detergent will not stably associate (lane f, and *Figure 7B and C*). Since there is no fusion without added sQa (*Figure 8A*, curve c), the corresponding complex

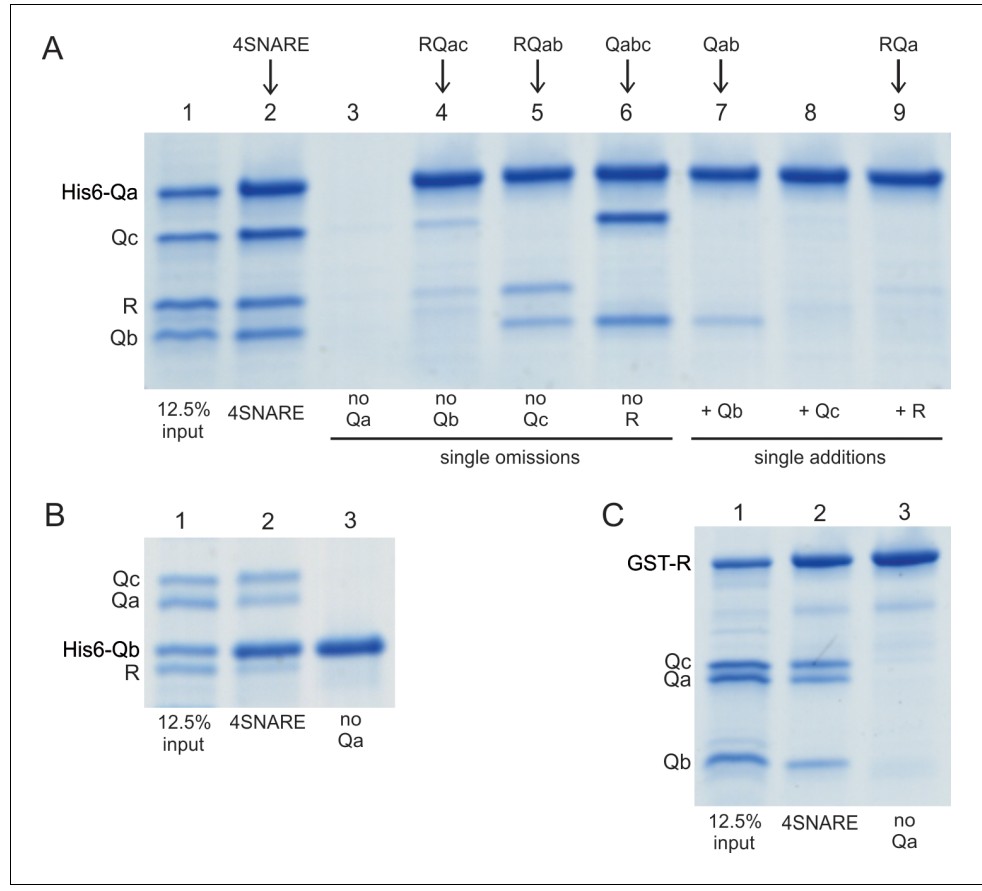

**Figure 7.** Spontaneous formation of SNARE complexes in detergent. His$_6$-tagged Qa SNARE (**A**), his$_6$-tagged Qb SNARE (**B**) or GST-tagged R SNARE (**C**) were mixed at 4 µM with 4 µM of the other indicated full-length SNAREs, in a total volume of 50 µl in pulldown buffer (20 mM HEPES-NaOH, pH 7.4, 150 mM NaCl, 10% glycerol, 100 mM ß-octylglucoside), plus 20 mM imidazoleCl, pH 7.0 for incubations with a his$_6$-tagged SNARE. After nutation for 1 hr at 4 ˚C, a portion (40 µl) was transferred to tubes containing either 20 µl of a 50% slurry of (**A, B**) nickel-NTA agarose (Qiagen, Hilden, Germany) or (**C**) glutathione agarose 4B (Genesee Scientific, San Diego, CA). Each was nutated at 4 ˚C for 1 hr, diluted with 0.5mls of pulldown buffer (**C**) or pulldown buffer plus imidazole (**A, B**), and centrifuged (500xg, 6 min, 4 ˚C). Supernatants were removed, and the beads were washed three more times with 0.5 ml portions of the same buffer. Proteins were eluted with 50 µl of SDS sample buffer with ß-mercaptoethanol by heating (95˚C, 5 min). Eluates were analyzed by Coomassie-stained gel. The substantial increase in molecular weight for his$_6$-Qa and his$_6$-Qb is caused by the presence of a 36 amino acyl linker between the his$_6$ tag and the N-terminus of each of these SNAREs (*Izawa et al., 2012*).

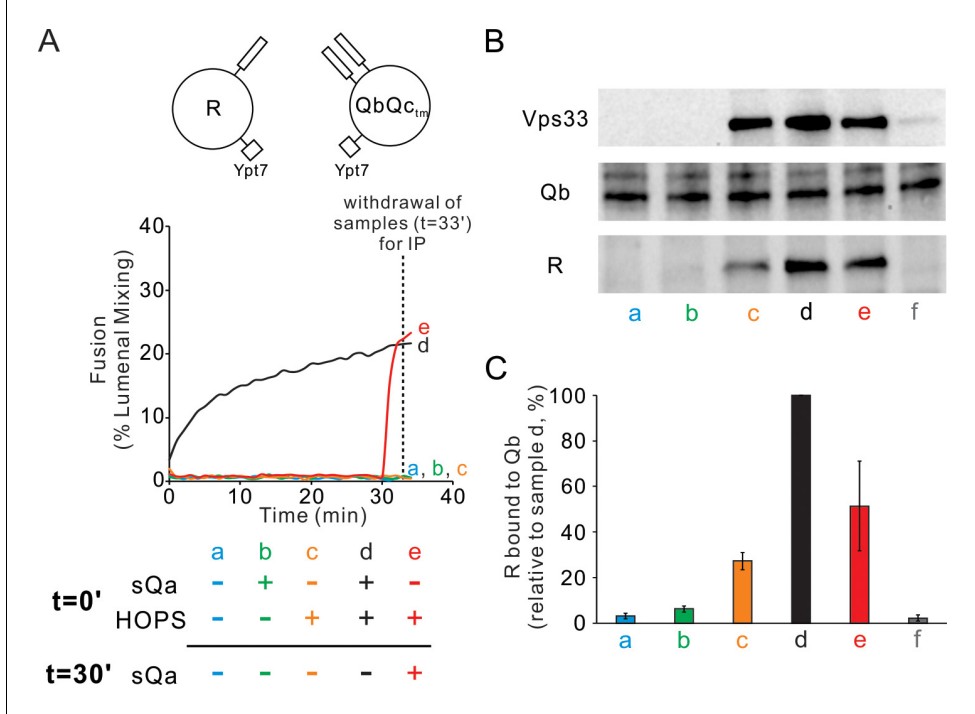

**Figure 8.** HOPS mediates the assembly of R- and QbQc$_{tm}$- SNAREs into a shared *trans*-complex in the absence of Qa. Mixed proteoliposomes bearing Ypt7 (1:8000 molar ratio to lipids) and either R- or QbQc$_{tm}$-SNAREs (1:16,000 molar ratio to lipids) were mixed with 50 nM HOPS and 4 μM sQa where indicated, added either at the start or after 30-min incubation. (A) Fusion was measured by FRET signal. (B) Samples were withdrawn at 33 min, solubilized in RIPA buffer, and the R SNARE that co-immunoprecipitated with 1.25 μg anti-Qb antibody was assayed as a measure of *trans*-complex, as described in *Figure 3* with 1.25 μg of antibody to Qb. (C) The average and standard deviation of the Nyv1 band intensity from three independent experiments are shown, normalized to sample d (HOPS and sQa added at t = 0).

The online version of this article includes the following source data for figure 8:

**Source data 1.** Source data file (Excel) for *Figure 8A and C*.

---

(*Figure 8B and C*, lane c) is all in trans. In contrast, at least some of the complex seen when sQa had been present during the 30 min incubation (*Figure 8B and C*, lane d) may be *cis*-complex that had assembled from SNAREs brought together by fusion (*Figure 8A*, lane d), as there was less complex seen just 2 min after sQa addition (*Figure 8B and C*, lane e vs d), although they had undergone comparable fusion (*Figure 8A*, curve e vs d). Furthermore, the HOPS-mediated rapid-fusion intermediate formed between QbQc and R is stable in the presence of Sec17, Sec18, and ATP (*Figure 4B*), whether present from the start of incubation or added after 25 min (d, no Sec17 or Sec18; e, addition from t = 0; f, added after 25 min incubation), as seen for HOPS-mediated intermediates with QaQb and R (*Figure 4A*).

Will membrane-anchored Qb suffice without Qc for HOPS-dependent assembly of a rapid-fusion complex with R-proteoliposomes? Proteoliposomes with Ypt7 and R fused with proteoliposomes bearing Ypt7 and the Qb and Qc-$_{tm}$ SNAREs when given HOPS and sQa (*Figure 9A*, solid black curve a) at a comparable rate to that seen with Ypt7/Qb proteoliposomes in the presence of HOPS, sQa, and Qc (dotted black curve c). Preincubation of HOPS with Ypt7/R- and Ypt7-QbQc$_{tm}$ proteoliposomes yielded substantially more rapid fusion upon sQa addition (solid red curve b) than when sQa had been added from the start (curve a), as shown above (*Figures 6A* and *8*). However, HOPS incubation with Ypt7/R-proteoliposomes and Ypt7/Qb proteoliposomes did not yield a rapid-fusion intermediate (*Figure 9A*, dotted green curve f), and inclusion of either Qc or sQa from the start of the incubation of HOPS with Ypt7/R- and Ypt7/Qb- proteoliposomes did not markedly enhance the rate of fusion upon the later addition of sQa or Qc, respectively (dotted red and blue curves d, e). The analogous comparison was done between fusion reactions which included Ypt7/R

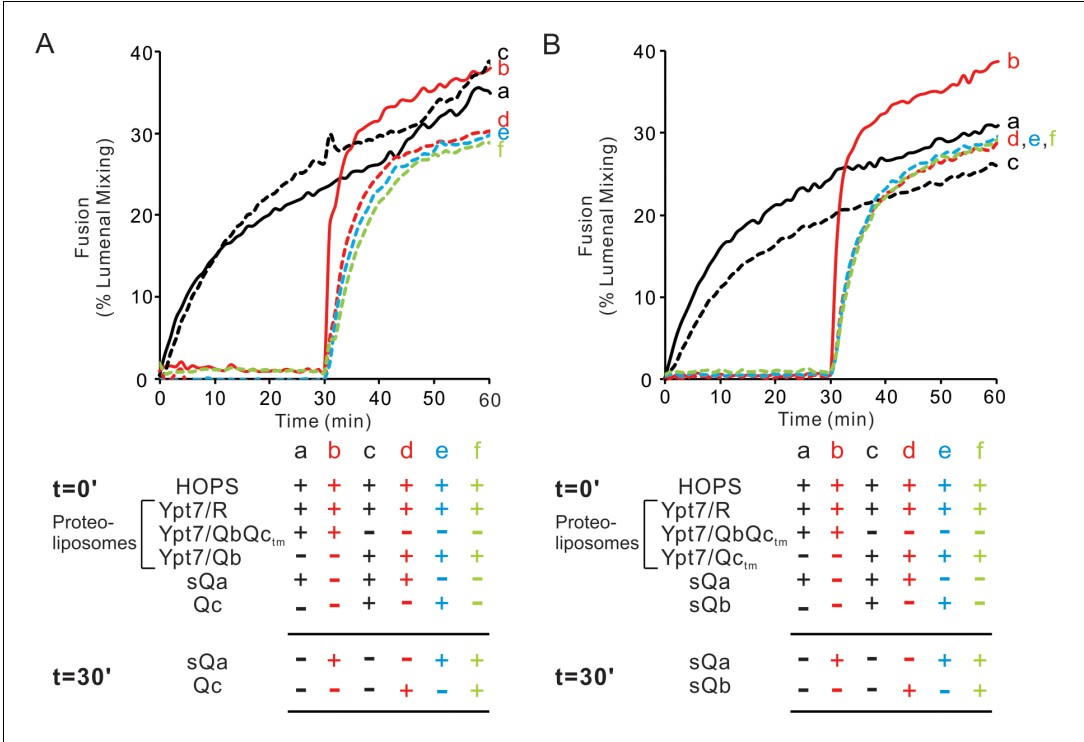

**Figure 9.** Both the Qb- and Qc-SNAREs are needed for the assembly of a rapid fusion intermediate without Qa. (A) Fusion reactions had mixed proteoliposomes bearing the R-SNARE and either Qbc_tm (lines) or Qb (dotted lines)-SNAREs (1:8000 molar ratio to lipids). These were mixed with 50 nM HOPS at t = 0. Also at t = 0, soluble SNAREs were added: sQa (a,c,d) and/or Qc (c and e). After 30 min, sQa (b, e, f) and Qc (d, f) were added as indicated in the reaction scheme. (B) Fusion reactions had proteoliposomes bearing R-SNARE and proteoliposomes with either Qbc_tm (solid lines) or Qc_tm (dotted lines)-SNAREs (1:8000 molar ratio to lipids). These were mixed with 50 nM HOPS at t = 0. Soluble SNAREs were also added at t = 0 as indicated: sQa (a, c, d) and sQb (c, e). After 30 min, sQa (b, e, f) and sQb (d, f) or sQa and sQb were added. All proteoliposomes had Ypt7-tm at a 1:8000 protein:lipid molar ratio. Content mixing assays in this figure are representative of n ≥ 3 experiments; means and standard deviations from four independent experiments are in *Figure 9—figure supplement 1*. The online version of this article includes the following source data and figure supplement(s) for figure 9:

**Source data 1.** Source data file (Excel) for *Figure 9A and B*.
**Figure supplement 1.** The assembly of a rapid-fusion intermediate without Qa needs membrane-bound Qb and Qc SNAREs.

proteoliposomes and either Ypt7/QbQc-tm proteoliposomes or Ypt7/Qc-tm proteoliposomes (*Figure 9B*). The rapid-fusion state formed by HOPS, Ypt7/R proteoliposomes, and Ypt7/QbQc-tm proteoliposomes (*Figure 9B*, red curve b) was not seen when Ypt7/R and Ypt7/Qc-tm proteoliposomes were incubated with HOPS, either alone (green curve f) or with sQb (blue curve e) or sQa (dotted red curve d), for 30 min prior to addition of the missing soluble Q-SNAREs. Thus, while HOPS can stabilize a rapid-fusion intermediate between R and Qa alone without Qb or Qc (*Figure 5*), both Qb and Qc-tm are needed for the accumulation of rapid-fusion complex in the absence of Qa (*Figure 9*).

## HOPS function is saturable for each Q-SNARE

We complemented our physical assays of HOPS binding to single SNAREs (*Figure 2*) and the capacity of HOPS to promote physical associations between R- and Q-SNAREs (*Figures 3* and *8*) which correspond to rapid-fusion intermediates (*Figure 6*) with assays of whether the HOPS-dependent fusion between proteoliposomes bearing Ypt7 plus the R-SNARE and those with Ypt7 plus a single integrally-anchored Q-SNARE was saturable at low concentrations of each Q-SNARE, a hallmark of

active site catalysis (*Figure 10*). The fusion of proteoliposomes that have Ypt7, R-SNARE, and lumenally entrapped biotinylated phycoerythrin with those bearing Ypt7, Qa-SNARE, and lumenally entrapped Cy5-streptavidin is supported by sQb, Qc, and an additional agent, either HOPS (*Song and Wickner, 2017*) or polyethylene glycol (PEG). While HOPS can specifically bind SNAREs (*Figure 2*), PEG is a nonspecific dehydrating agent (*Lentz, 2007*) which clusters membranes and

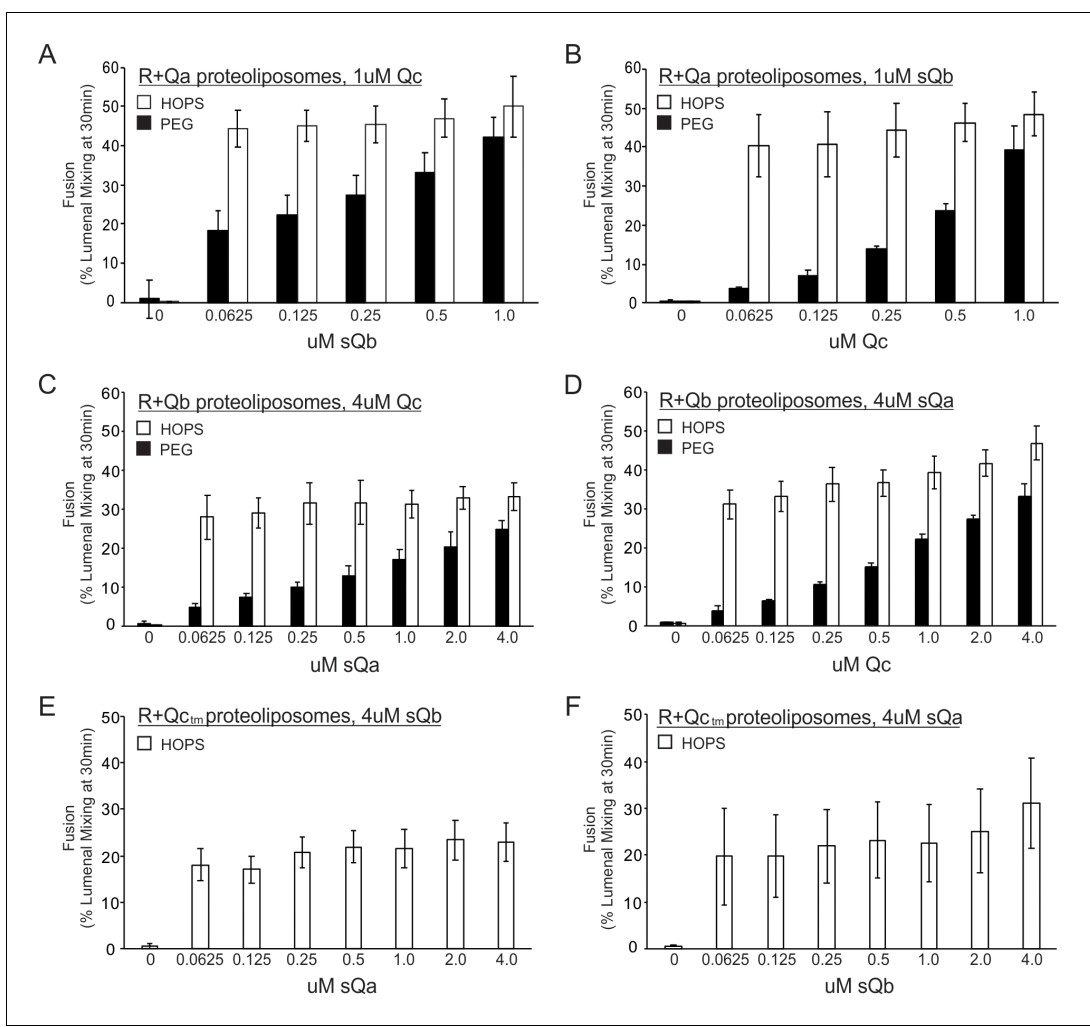

**Figure 10.** Fusion with HOPS is saturable for each vacuolar Q-SNARE. Reconstituted proteoliposomes of VML composition were prepared with wild-type Ypt7 at 1:4000 protein to lipid molar ratio and either R-SNARE or each single Q-SNARE at a 1:2500 protein to lipid ratio, employing a transmembrane version of Qc. Fusion assays were performed in RB150. Ypt7/R-SNARE proteoliposomes and Ypt7/Q-SNARE proteoliposomes were separately incubated at 1 mM (lipid) with 20 μM streptavidin, 2 mM EDTA, 0.5 mM MgCl$_2$, and 1 mM GTP for 10 min at 27 °C. MgCl$_2$ was then added to bring the concentration to 2.5 mM. The nucleotide-exchanged R- and Q-proteoliposomes were then combined and portions were added to tubes containing one half volume of either 0.16 μM HOPS or 8% PEG. Aliquots of each (16 μl) were pipetted into a 384-well plate. During the nucleotide exchange process, a mixture of the missing soluble Q-SNAREs was prepared in RB150, containing 4 μM of each soluble Q-SNARE (**A and B**) or 16 μM of each soluble Q-SNARE (**C**), (**D**), (**E**), and (**F**). Two dilution curves were then prepared, keeping one soluble SNARE at the starting concentration while diluting the other twofold. A portion (5 μl) of each dilution was pipetted into empty wells of a 384-well plate, which then received 15 μl of the mixtures of proteoliposome with HOPS or PEG. Final concentrations of HOPS or PEG in the 20 μl reaction were 40 nM and 2%, respectively.

The online version of this article includes the following source data and figure supplement(s) for figure 10:

**Source data 1.** Source data file (Excel) for *Figure 10A,B,C,D,E and F*.

**Figure supplement 1.** An MBP tag on the soluble Qb-SNARE interferes with its recruitment by HOPS.

promotes SNARE assembly without any SNARE-binding specificity. With PEG, the fusion rate steadily declines with diminishing sQb (*Figure 10A*, filled bars), as expected for four SNAREs spontaneously assembling into a required tetramer. However, with HOPS the rate is almost constant over the same wide sQb concentration range (*Figure 10A*, open bars), indicating saturation of an active HOPS binding site for sQb. [Earlier studies of HOPS-mediated fusion between R- and Qa-SNARE proteoliposomes had employed MBP-sQb, and found that it hadn't exhibited saturable kinetics (*Zick and Wickner, 2013*). We reproduce this finding (*Figure 10—figure supplement 1*, filled bars), and note that the MBP 'tag' had prevented a high-affinity, saturable engagement with HOPS, which is seen upon proteolytic removal of the tag (open bars).] When Ypt7/R and Ypt7/Qa proteoliposomes were mixed with ample sQb and the concentration of Qc was varied, fusion with PEG tethering was again proportional to the Qc concentration, while fusion with HOPS as the tether showed little change over a wide range of Qc (*Figure 10B*). This saturation indicates that HOPS has a functional Qc binding site. In a similar approach, proteoliposomes bearing Ypt7 and R were incubated with those bearing Ypt7 and Qb, in the presence of sQa, Qc, and either HOPS or PEG. With HOPS, the rate of fusion was saturable with respect to the concentration of Qa or Qc (*Figure 10C and D*). HOPS-dependent fusion of Ypt7/R and Ypt7/Qc proteoliposomes, where Qc was fused to a membrane anchor, is also invariant over a wide range of sQa or sQb concentrations (*Figure 10E and F*); direct comparison with PEG-mediated fusion was not possible, as PEG did not support the SNARE-dependent fusion of these proteoliposomes. In short, only tethered proteoliposomes will assemble *trans*-SNARE complexes and proceed to fuse. Once HOPS or PEG has tethered the membranes, *trans*-SNARE assembly can begin. If HOPS had no function beyond tethering, then membranes tethered by HOPS or PEG would have the same Km for each SNARE. However, when HOPS, which can recognize each SNARE, performs the tethering, we find that fusion has a far lower Km for each SNARE than when tethering is through PEG, which cannot recognize SNAREs. This indicates that HOPS not only functions by tethering but also by its recognition of each individual SNARE. The ability of HOPS to bind each SNARE (*Figure 2*), the low Km saturability of HOPS-mediated fusion for each Q-SNARE (*Figure 10*) and the capacity of HOPS to assemble a rapid-fusion complex between R- and Q-SNAREs in trans (*Figures 3–6* and *8*) demonstrate a central role of HOPS in the recognition of each Q-SNARE and in the assembly of rapid-fusion intermediates.

## Discussion

The proteins and lipids which mediate homotypic vacuole fusion cluster around the edge of the apposed membranes of docked vacuoles (*Wang et al., 2002*) and are interdependent for this localization (*Fratti et al., 2004*). The multiplicity of affinities among these proteins is striking, and may underlie the interdependent character of their microdomain enrichment and functions for fusion. For example, HOPS binds each of the 4 SNAREs (*Stroupe et al., 2006*; *Baker et al., 2015*; *Figure 2*), acidic lipids (*Karunakaran and Wickner, 2013*), phosphoinositides (*Stroupe et al., 2006*), and Ypt7 on each of 2 docked membranes (*Hickey and Wickner, 2010*). While multiple binding affinities are perhaps expected for a large, multi-subunit complex such as HOPS, even Sec17 binds to SNAREs (*Söllner et al., 1993*; *Zick et al., 2015*), lipid (*Zhao et al., 2015*; *Zick et al., 2015*), and Sec18 (*Weidman et al., 1989*). Each of these components is required for fusion, both in vivo and in vitro with purified vacuoles. The purification of each of these proteins allows exploration of their mutual affinities, while the creation of natural and synthetic fusion sub-reactions allows tests of their functionality.

Why is HOPS needed for membrane fusion, and what does it do? HOPS provides tethering (*Hickey and Wickner, 2010*) through the Ypt7 affinities of its Vps39 and 41 subunits (*Brett et al., 2008*), and tethering per se suffices without SNARE recognition for efficient fusion once the Q-SNAREs are assembled (*Figure 1*; *Song and Wickner, 2019*). Q-SNARE assembly can be catalyzed by HOPS. In an earlier model sub-reaction (*Orr et al., 2017*), proteoliposomes bearing Ypt7 and R-SNARE were incubated with the three soluble Q-SNAREs and HOPS, then re-isolated by floatation. HOPS was required for the association of each of the Q-SNAREs, and each Q-SNARE depended on the other two for its HOPS-dependent membrane association. This is one means of assaying HOPS-dependent 4-SNARE complex assembly, albeit in cis. While HOPS supports the assembly of those *cis*-complexes, dependent on all 4 SNAREs, *cis*-complexes with 3 SNAREs were not stable. We now report that HOPS recognizes each SNARE (*Figure 2*), and show that these

recognitions support the assembly of HOPS:R:Qa and HOPS:R:QbQc rapid-fusion *trans*-SNARE intermediates. The affinity of HOPS for membrane-anchored Qb was overlooked in earlier studies (*Stroupe et al., 2006*) because it may be the product of two affinities, a modest (e.g. micromolar) affinity of HOPS for Qb for and a low affinity (e.g. millimolar) for the lipid bilayer, yielding together a high (e.g. nanomolar) affinity. In our current study, we examine intermediates in trans-SNARE assembly which lack one or the other of the 3 Q-SNAREs and report the existence of rapid-fusion intermediates for each of the 3 'missing' Q-SNAREs, including Qa. Single-molecule force microscopy has been used in an elegant demonstration that the HOPS Vps33 SM subunit can template 4-SNARE complex assembly through association with covalently-joined R- and Qa-SNAREs (*Jiao et al., 2018*). Our current studies show that when a proteoliposomal Q-SNARE fusion partner has only two bound Q-SNAREs instead of all three and the third Q-SNARE is present in soluble form, HOPS is essential for fusion, and GST-PX will not suffice (*Figure 1*). HOPS catalyzes the entry of each Q-SNARE into complex which is poised for rapid fusion (*Figures 3–6*, *8* and *9*).

Our current findings place SM protein recognition of the SNARE domains of R- and Qa-SNAREs in the context of recruitment of each of the 4 SNAREs. With the discovery (*Baker et al., 2015*) of conserved grooves on the surface of the HOPS SM-family subunit Vps33 which bind the R- and Qa-SNARE domains in parallel (N to C) and in register (with adjacent 0-layer residues), it was possible that these associations are a unique and committed step for 4-SNARE assembly. One limitation to the concept that R and Qa can only associate during templating by an SM protein is that 4-SNARE assembly of R- with Q-SNAREs can proceed without SM function as long as there is tethering and the three Q-SNAREs are pre-assembled (*Figure 1A and B*). In this context, the Qb and Qc SNAREs which are associated with Qa may substitute for the SM templating function. It is unclear whether the three Q-SNAREs ever physiologically pre-assemble in the presence of Sec17, Sec18, and ATP.

HOPS has the unique capacity to create a rapid-fusion intermediate of proteoliposomes bearing R-SNARE with those bearing any two Q-SNAREs, able to receive the third Q-SNARE for rapid fusion (*Figure 6*), and a mere tether will not suffice (*Figure 1C–E*). Presumably, the assembly of this intermediate requires the R-SNARE binding site of the HOPS Vps33 SM-subunit and as well as binding sites for Qa on Vps33 or for Qb and Qc on other HOPS subunits. We find HOPS-dependent assembly of a rapid-fusion intermediate which includes the Qb and R-SNAREs in the absence of Qa (*Figures 6A* and *8*), even though these SNAREs by themselves cannot associate stably (*Figure 7*). The R- and Qb-SNAREs have minimal contacts in a 4-SNARE complex (*Sutton et al., 1998*) which may explain the need for Qc as well as Qb for this rapid-fusion intermediate (*Figure 9*). Our compositional analysis in detergent extracts shows that these intermediates include both the R- and Q-SNAREs which were in trans, whether these SNAREs are directly associated with each other in an incomplete SNARE coiled coils bundle or are only associated through the binding of each to their respective binding sites on the HOPS complex. While the precise composition and structure of these activated complexes will be of great interest, it will also be a major technological challenge. Only a few per cent of the SNAREs are engaged in trans-associations at any time (*Collins and Wickner, 2007*) and HOPS has many binding affinities, for the Rab (*Seals et al., 2000*), the SNAREs (*Figure 4*, and *Stroupe et al., 2006*; *Baker et al., 2015*), and specific lipids (*Stroupe et al., 2006*; *Karunakaran and Wickner, 2013*). Since a small proportion of the SNAREs and HOPS are engaged to activate membranes for rapid fusion, and these structures span two apposed bilayers, assaying their detailed composition and conformations will be challenging.

We suggest a working model (*Figure 11*). The binding sites on HOPS for each of the four individual SNAREs mediate the initial HOPS:SNARE associations (Step A). If the initial SNAREs to associate are R and Qa (left), the apolar surfaces of their alpha helices, which initially face into their respective grooves on Vps33, may be released to turn toward each other while the nascent R:Qa *trans*-complex remains stabilized in association with HOPS through some low affinities of HOPS for their N-domains or the polar surfaces of their SNARE domains (Step B, left). Similarly, for the HOPS:RQbQc intermediate, each of these 3 SNAREs initially associate with their individual HOPS binding sites (Step A, right), but then may associate with each other in a ternary coiled-coils complex which is stabilized by modest-affinity HOPS association with their N-domains or polar surfaces of their SNARE domains (Step B, right). In contrast, HOPS:R:Qb (without Qa or Qc) or HOPS:R:Qc (without Qa or Qb) are not sufficiently stable to accumulate as rapid-fusion intermediates in the strained configuration of being anchored to two membranes. Each intermediate, whether HOPS:R:Qa [alone or with Qb or Qc] or

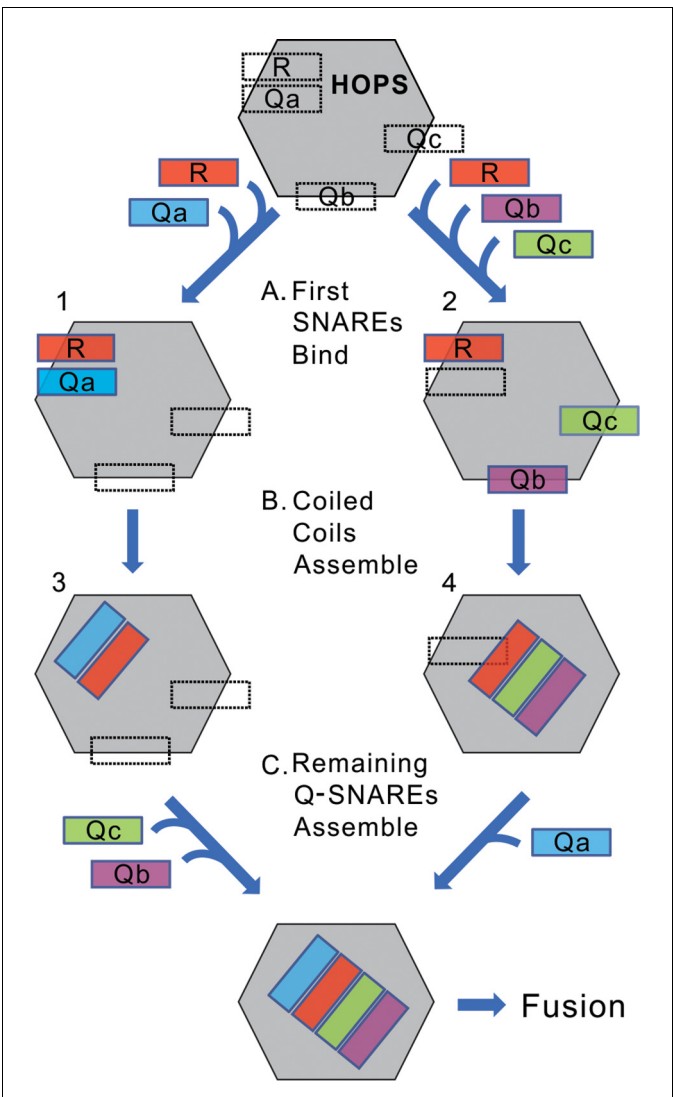

**Figure 11.** A conceptual model of rapid-fusion intermediates. HOPS has binding sites for each of the four vacuolar SNAREs, indicated in dotted lines. We propose that (**A**) R, and either Qa (left) or Qb and Qc (right), bind to HOPS at their high-affinity sites, then (**B**) partially or wholly reorient to begin their coiled-coils assembly. (**C**) The binding sites for the remaining Q-SNAREs catalyze their rapid transfer to the nascent coiled-coil, triggering rapid fusion. Our data do not establish when SNAREs leave their initial binding sites to begin coiled-coils association; the rapid-fusion intermediates might be represented by 1 and 2, or 3 and 4, or all 4 SNAREs might remain bound to their initial sites before switching to coiled-coils association.

HOPS:R:Qb:Qc, and whether the SNAREs remain bound to their initial sites on HOPS or have begun coiled-coils assembly, is poised to accept the missing SNAREs (Step C) for very rapid fusion.

It remains unclear whether all the components needed for fusion remain engaged with each other up to and during lipid bilayer mixing; is there a 2Ypt7/HOPS/4SNARE/2Sec17/Sec18 complex? Intermediates such as HOPS-mediated *trans*-association of R and Qa with Qc can be labile (*Figure 5B*) yet functionally important. Lability may derive from the strain on the SNARE complex imposed by its anchors to tightly apposed and bent bilayers. Earlier studies of *cis*-SNARE complexes from isolated vacuoles showed that HOPS and Sec17 were in separate complexes with SNAREs, and suggested that Sec17 could displace HOPS from SNARE associations (*Collins et al., 2005*). It is also unclear whether HOPS remains bound to Ypt7 and even whether it remains bound to the SNAREs. We have noted (*Baker et al., 2015*) that the helical R and Qa SNARE domains bind to their sites on Vps33, the HOPS SM-family subunit, with the same face of the SNARE domain helix in contact with Vps33

as faces inward toward the other SNAREs in the assembled 4-helical SNARE complex (*Sutton et al., 1998*). These SNAREs are thus likely to leave their Vps33 contact sometime prior to completion of SNARE zippering, though the whole SNARE complex may exploit the HOPS affinities for the Qb and Qc SNAREs to remain bound.

The binding grooves for the R- and Qa-SNARE domains are conserved (*Baker et al., 2015*), suggesting a model of templating for SNARE complex formation at other organelles. Recombinant vacuolar and neuronal SM proteins have been shown by single molecule force spectroscopy to mediate SNARE assembly (*Jiao et al., 2018*). It remains unclear whether other proteins involved in fusion at other organelles directly recognize their Qb and Qc SNAREs, catalyzing their entry into SNARE complexes as reported here for HOPS.

The vacuole fusion reaction has been studied in cells, with the isolated organelle, and with purified components reconstituted into proteoliposomes. The latter approach allows reconstitution and assay of subreactions, addressing mechanistic questions and testing and revising models. In early models of fusion, tethering simply provided SNARE proximity to each other for spontaneous *trans*-SNARE complex assembly. SNAREs then zippered spontaneously, distorting the bilayers for fusion. After fusion, SNARE NSF/Sec18 and αSNAP/Sec17 function as an ATP-driven SNARE disassembly chaperone system to disassemble *cis*-SNARE complexes for the subsequent round of fusion. Recent studies have refined this model. Tethering brings all the fusion proteins and lipids into proximity, allowing an interdependent enrichment in a dedicated fusion microdomain. Membrane tethering is needed for SNAREs to assemble in trans in a fusion-competent conformation (*Song and Wickner, 2019*). Large Rab-effector complexes, such as vacuolar/lysosomal HOPS, will mediate tethering (*Baker and Hughson, 2016*) and guide and catalyze SNARE complex assembly. HOPS also coordinates the loading of Sec17 and Sec18 onto assembled SNAREs, and these chaperones may promote fusion by some combination of adding wedge-like bulk to the fusion domain (*D'Agostino et al., 2017*), promoting SNARE zippering (*Song et al., 2017*), and distorting bilayers adjacent to the SNAREs with the Sec17 apolar loop (ibid). It remains unclear whether HOPS and Sec17 remain associated with *trans*-SNARE complexes at the same time and how Sec18 can contribute to fusion without disassembling the *trans*-SNARE complexes.

# Materials and methods

**Key resources table**

| Reagent type (species) or resource | Designation | Source or reference | Identifiers | Additional information |
|---|---|---|---|---|
| Gene (*Saccharomyces cerevisiae*) | Nyv1 | Saccharomyces Genome Database | SGD:S000004083 | |
| Gene (*Saccharomyces cerevisiae*) | Vam3 | Saccharomyces Genome Database | SGD:S000005632 | |
| Gene (*Saccharomyces cerevisiae*) | Vti1 | Saccharomyces Genome Database | SGD:S000004810 | |
| Gene (*Saccharomyces cerevisiae*) | Vam7 | Saccharomyces Genome Database | SGD:S000003180 | |
| Gene (*Saccharomyces cerevisiae*) | Ypt7 | Saccharomyces Genome Database | SGD:S000004460 | |
| Gene (*Saccharomyces cerevisiae*) | Sec17 | Saccharomyces Genome Database | SGD:S000000146 | |
| Gene (*Saccharomyces cerevisiae*) | Sec18 | Saccharomyces Genome Database | SGD:S000000284 | |

*Continued on next page*

*Continued*

| Reagent type (species) or resource | Designation | Source or reference | Identifiers | Additional information |
|---|---|---|---|---|
| Peptide, recombinant protein | GST-R (Nyv1) | PMID: 18650938 | | purified from *E. coli.* |
| Peptide, recombinant protein | GST-Qa (Vam3) | PMID: 18650938 | | purified from *E. coli.* |
| Peptide, recombinant protein | GST-Qb (Vti1) | PMID: 18650938 | | purified from *E. coli.* |
| Peptide, recombinant protein | GST-sR (soluble) | PMID: 15241469 | | purified from *E. coli.* |
| peptide, recombinant protein | GST-sQa (soluble) | PMID: 28637767 | | purified from *E. coli.* |
| Peptide, recombinant protein | MBP-sQb (soluble) | PMID: 24088569 | | purified from *E. coli.* |
| Peptide, recombinant protein | Vam7-tm | PMID: 23071309 | | purified from *E. coli.* |
| Peptide, recombinant protein | Ypt7-tm | PMID: 31235584 | | purified from *E. coli.* |
| Peptide, recombinant protein | $His_6$-Qa | PMID: 22174414 | | purified from *E. coli.* |
| Peptide, recombinant protein | $his_6$-Qb | PMID: 22174414 | | purified from *E. coli.* |
| Peptide, recombinant protein | Vam7 | PMID: 17699614 | | purified from *E. coli.* |
| Peptide, recombinant protein | TEV protease | PMID: 18007597 | | purified from *E. coli.* |
| Peptide, recombinant protein | HOPS | PMID: 18385512 | | purified from *Saccharomyces cerevisiae.* |
| Peptide, recombinant protein | GST-PX | PMID: 23071309 | | purified from *E. coli.* |
| Antibody | anti-Vam3 (rabbit polyclonal) | PMID: 12566429 | Wickner lab stock | WB: 0.67 µg/ml IP: 5 µg |
| Antibody | anti-Nyv1 (rabbit polyclonal) | PMID: 10385523 | Wickner lab stock | WB: 1 µg/ml |
| Antibody | anti-Vti1 (rabbit polyclonal) | PMID: 18007597 | Wickner lab stock | WB: 2 µg/ml IP: 1.25 µg |
| Antibody | anti-Vps16 (rabbit polyclonal) | PMID: 18007597 | Wickner lab stock | WB: 0.92 µg/ml |

*Continued*

| Reagent type (species) or resource | Designation | Source or reference | Identifiers | Additional information |
|---|---|---|---|---|
| Antibody | anti-Vps33 (rabbit polyclonal) | PMID: 10944212 | Wickner lab stock | WB: 0.5 μg/ml |
| Chemical compound, drug | Cy5-derivatized streptavidin | SeraCare Life Sciences | 5270–0023 | |
| Chemical compound, drug | Biotinylated PhycoE | Thermo Fisher Scientific | p811 | |
| Chemical compound, drug | streptavidin | Thermo Fisher Scientific | 434302 | |
| Chemical compound, drug | 1,2-dilinoleoyl-sn-glycero-3-phosphocholine | Avanti polar lipids | 850385 | |
| Chemical compound, drug | 1,2-dilinoleoyl-sn-glycero-3-phospho-L-serine | Avanti polar lipids | 840040 | |
| Chemical compound, drug | 1,2-dilinoleoyl-sn-glycero-3-phosphoethanolamine | Avanti polar lipids | 850755 | |
| Chemical compound, drug | 1,2-dilinoleoyl-sn-glycero-3-phosphate | Avanti polar lipids | 840885 | |
| Chemical compound, drug | L-α-phosphatidylinositol | Avanti polar lipids | 840044 | |
| Chemical compound, drug | 1,2-dipalmitoyl-sn-glycerol | Avanti polar lipids | 800816 | |
| Chemical compound, drug | ergosterol | Sigma | 45480 | |
| Chemical compound, drug | PI(3)P diC16 | Echelon Bioscience | P-3016 | |
| Chemical compound, drug | rhodamine DHPE | Invitrogen | L1392 | |
| Chemical compound, drug | NBD-PE | Invitrogen | N360 | |
| Chemical compound, drug | Marina-blue | Invitrogen | M12652 | |
| Software and Algorithms | UN-SCAN-IT | Silk Scientific | | |

## Proteins and reagents

The soluble version of GST-Qa (GST-sQa), with Qa amino acyl residues 1–264 but lacking its transmembrane domain, was generated by PCR with the Phusion high-fidelity DNA polymerase (NEB). The DNA fragment was cloned into BamHI and SalI digested pGST parallel1 vector (*Sheffield et al., 1999*) with an in-Fusion kit (Clonetech).

For GST-sVam3,
F: AGGGCGCCATGGATCCGATGTCCTTTTTCGACATCGA
R: AGTTGAGCTCGTCGACTACTTACCGCATTTGTTACGGT

Full-length, untagged Nyv1 (Mima et a.l, 2008) was cloned into BamHI and SalI digested pGST parallel1 vector (*Sheffield et al., 1999*) with the HiFi DNA assembly kit (New England Biolabs, Ipswich, MA).

For GST-Nyv1:

F: AGGGCGCCATGGATCCGATGAAACGCTTTAATGTAAGT

R: AGTTGAGCTCGTCGATTACCACAGATAGAAAAACAT

Trans-membrane (tm)-anchored Ypt7: The nucleotide sequence encoding the transmembrane domain of the Qa-SNARE Vam3 (amino acyl residues 265–283) fused to the 3′ end of the nucleotide sequence encoding full length Ypt7 was amplified by PCR from pET-19 Ypt7-tm (a kind gift from C Ungermann) with the Phusion high-fidelity DNA polymerase (NEB). The DNA fragment was cloned into BamHI and SalI digested pMBP-parallel1 vector (*Sheffield et al., 1999*) with the HiFi DNA assembly kit (New England Biolabs, Ipswich, MA).

For Ypt7-tm

F: AGGGCGCCATGGATCCGTCTTCTAGAAAAAAAAATATTTT

R: AGTTGAGCTCGTCGACTAACTTAATACAGCAAGCA

The resulting plasmid sequence was confirmed.

The purifications of HOPS (*Zick and Wickner, 2013*), GST-PX (*Fratti et al., 2004*), Sec17p (*Schwartz and Merz, 2009*), Sec18p (*Mayer et al., 1996*), wild-type Ypt7 (*Zick and Wickner, 2013*), and a soluble version of MBP-Qb (MBP-sQb) lacking its transmembrane domain (*Zick and Wickner, 2013*) were as described. Full-length, wild-type vacuolar SNAREs GST-Qa, Qc, R, and Qb were isolated as described (*Mima et al., 2008*; *Schwartz and Merz, 2009*; *Zucchi and Zick, 2011*), and Qb and R were buffer exchanged into β-octylglucoside (*Zucchi and Zick, 2011*). Vam7-tm (*Xu and Wickner, 2012*) and Sec17-tm (*Song et al., 2017*) were purified as described. The plasmid encoding his$_6$-Vam3 (full length) and his$_6$-Vti1 (full length) were kind gifts from Joji Mima, and the protein was purified as described (*Izawa et al., 2012*).

GST-Nyv1 and MBP-Ypt7-tm were purified as follows: GST-Nyv1 and MBP-Ypt7-tm were produced in *E. coli* Rosetta(DE3)*pLysS* (Novagen, Milwaukee WI). A single colony was inoculated into 50 ml LB medium containing 100 µg/ml ampicillin (Amp) and 37 µg/ml Chloramphenicol (Cam) and grown overnight at 37°C, then transferred to 6 l LB with 100 µg/ml Amp and 37 µg/ml Cam. Cultures were grown at 37°C to an OD$_{600}$ of 0.5. IPTG (0.5 mM) was added and cultures were shaken for 3 hr at 37°C. Cells were harvested by centrifugation (Beckman JA10 rotor, 5000 rpm, 5 min, 4°C) and resuspended in 50 ml buffer A (20 mM HEPES/NaOH, pH 7.4, 100 mM NaCl, 1 mM EDTA, 1 mM DTT, 1 mM PMSF [phenylmethylsulfonyl fluoride] and PIC [protease inhibitor cocktail; *Xu and Wickner, 1996*]). Resuspended cells were lysed by French Press (8000 psi, 4°C, two passages) and lysates were centrifuged (Beckman 60Ti rotor, 30 min, 50,000 rpm, 4°C). Pellets were resuspended in 100 ml of buffer B (PBS [140 mM NaCl, 2.7 mM KCl, 10 mM Na$_2$HPO$_4$ and 1.8 mM KH$_2$PO$_4$, pH7.4], 1 mM EDTA, 1 mM dithiothreitol, 10% glycerol, PIC and 1 mM PMSF) with a Dounce homogenizer and centrifuged (60Ti, 50,000 rpm, 30 min, 4°C). Pellets were resuspended in 100 ml of buffer C (PBS, 1 mM EDTA, 1 mM DTT, 1% Triton X100, 10% glycerol, PIC and 1 mM PMSF) with a Dounce homogenizer and incubated (4°C) with nutation for 1 hr. The extract was centrifuged (60Ti, 50,000 rpm, 30 min, 4°C) and the supernatant was added to 24 mL of glutathione-Sepharose 4B resin for GST-Nyv1 (GE Healthcare, Pittsburg, PA) or 24 mL of amylose resin for MBP-Ypt7-tm (NEB, Ipswich MA) pre-equilibrated with buffer C and nutated for 2 hr at 4°C. The resin was gravity-packed into a 2.5 cm diameter column at 4°C, washed with 100 mL of buffer D (100 mM HEPES/NaOH, pH 7.4, 100 mM NaCl, 1 mM EDTA, 1 mM DTT, 100 mM β-OG, 10% glycerol). GST-Nyv1 was eluted with 40 mM reduced glutathione in buffer D and MBP-Ypt7-tm was eluted with 25 mM maltose in buffer D. Proteins were frozen in liquid nitrogen and stored at −80°C.

A plasmid encoding the soluble version of GST-R (GST-sR) lacking its transmembrane domain (*Thorngren et al., 2004*) was transformed into *E. coli* BL21(DE3) and the protein was purified as follows: 100 ml of LB+ 100 µg/ml Ampicillin was inoculated with a single colony, shaken overnight at 37°C, then added to 3L of LB+Ampicillin. Cultures were grown at 37°C to an OD$_{600}$ of 0.8, induced with 1 mM IPTG, and shaken overnight at 18°C. Cells were harvested and resuspended in 40mls resuspension buffer (20 mM TrisCl, pH 8.0, 200 mM NaCl, 200 µM PMSF, PIC). Cells were lysed by French Press (two passages) and lysates were centrifuged in a Beckman 60ti rotor (1 hr, 50,000 rpm, 4°C). The supernatant was nutated (2 hr, 4°C) with 10 ml glutathione-Sepharose 4B resin (GE Healthcare, Pittsburg, PA) in resuspension buffer. The slurry was poured into a column, the settled resin

was washed with resuspension buffer, and protein eluted with 100 mM HEPES-NaOH pH 7.8, 300 mM NaCl, 20 mM glutathione. The protein peak was dialyzed into RB150 (20 mM HEPES-NaOH pH 7.4, 150 mM NaCl, 10% glycerol [vol/vol]) in 6–8K molecular weight cutoff dialysis tubing (Fisher Scientific, Pittsburgh, PA), aliquoted, and frozen in liquid nitrogen. GST-sVam3 was purified the same way as GST-sNyv1, except that the growth media also contained 37 µg/ml chloramphenicol, the culture was grown to $OD_{600}$ of 1.0 before induction, the elution buffer was 20 mM HEPES-NaOH pH 7.4, 300 mM NaCl, 20 mM glutathione, 1 mM DTT, and the eluate was frozen in aliquots without dialysis. Before use, the MBP-sVti1, GST-sNyv1, and GST-sVam3 were cleaved with TEV protease to remove their tags, unless otherwise noted.

Dilinoleoyl lipids (diC18:2 PC, PS, PE, and PA), soy PI, and 1,2-dipalmitoyl-$sn$-glycerol were purchased from Avanti Polar Lipids (Alabaster, AL). Ergosterol was from Sigma Aldrich (St. Louis, MO), PI(3)P from Echelon Biosciences (Salt Lake City, UT), and the fluorescent lipids Marina-Blue DHPE, NBD-PE, and Lissamine rhodamine DHPE were from Invitrogen by Life Technologies (Eugene, OR). N-octyl-ß-D-glucopyranoside was from Anatrace (Maumee, OH), and poly(ethylene glycol) 8000 was from Sigma-Aldrich.

## Proteoliposome preparation

Proteoliposomes were prepared as described in *Zick and Wickner (2014)* with modifications. Lipid compositions of vacuolar mimic lipid (VML) proteoliposomes for content-mixing assays were 47.3 or 46.1 mol% diC18:2 PC, 18% diC18:2 PE, 18% soy PI, 4.4% diC18:2 PS, 2% diC18:2 PA, 8% ergosterol, 1% diacylglycerol, 1% diC16 PI(3)P and either 0.3% Marina Blue-PE or 1.5% NBD-PE. Lipid compositions of proteoliposomes for flotation assays were either 99% diC18:2 PC and 1% Lissamine rhodamine-DHPE or 83.5% diC18:2 PC, 15% diC18:2 PS and 1.5% NBD-PE. Proteins were added at protein:lipid ratios as described in the figure legends. Proteoliposomes were isolated by flotation through density medium as described (*Zick and Wickner, 2013*) and assayed for total phosphate (*Chen et al., 1956*). Aliquots of proteoliposomes in RB150+$Mg^{2+}$ (20 mM HEPES-NaOH, pH 7.4, 150 mM NaCl, 10% glycerol [vol/vol], 1 mM $MgCl_2$) were frozen in liquid nitrogen at a concentration of 2 mM lipid phosphorus.

## Fusion assay

Proteoliposomes were nucleotide exchanged by incubating proteoliposomes (1 mM lipid), RB150, streptavidin (10 µM), EDTA (2 mM), and GTP (20 µM) for 10 min at 27°C. Nucleotide exchange was completed by adding $MgCl_2$ (4 mM) and the mixture was placed on ice. After prewarming (10 min, 27°C), fusion was initiated by mixing 5 µl each of GTP exchanged R- and Q-SNARE proteoliposomes and adding soluble components (10 µL of for example, HOPS, GST-PX, and soluble SNAREs as noted), for a total volume of 20 µl. Plates (Corning 4514, 384 wells) were incubated at 27°C in SpectraMax Gemini XPS (Molecular Devices, Sunnyvale, CA) fluorescence plate reader and lumenal mixing was assayed every minute, as described (*Zick and Wickner, 2016*).

## Acknowledgements

We thank Michael Zick, Jose Rizo, Thomas Torng, Sarah Port, Gustav Lienhard, and Charles Barlowe for fruitful discussions, Christian Ungermann for the generous gift of a plasmid encoding Ypt7-tm, and Joji Mima for the kind gift of plasmids encoding his$_6$-tagged Qa and Qb. This work was supported by NIH grant R35GM118037. MH was supported by Deutsche Forschungsgemeinschaft fellowship HA 7730/2–1.

## Additional information

### Funding

| Funder | Grant reference number | Author |
| --- | --- | --- |
| National Institutes of Health | R35GM118037 | William T Wickner |
| Deutsche Forschungsgemeinschaft | HA 7730/2-1 | Max E Harner |

The funders had no role in study design, data collection and interpretation, or the decision to submit the work for publication.

### Author contributions
Hongki Song, Conceptualization, Resources, Data curation, Formal analysis, Validation, Investigation, Visualization, Methodology; Amy S Orr, Resources, Data curation, Formal analysis, Investigation, Methodology; Miriam Lee, Investigation, Methodology; Max E Harner, Resources, Methodology; William T Wickner, Conceptualization, Resources, Data curation, Supervision, Funding acquisition, Validation, Investigation, Methodology, Project administration

### Author ORCIDs
Hongki Song [ID] http://orcid.org/0000-0002-3761-5434
Max E Harner [ID] http://orcid.org/0000-0002-5513-1046
William T Wickner [ID] https://orcid.org/0000-0001-8431-0468

### Decision letter and Author response
Decision letter https://doi.org/10.7554/eLife.53559.sa1
Author response https://doi.org/10.7554/eLife.53559.sa2

# Additional files

### Supplementary files
• Transparent reporting form

### Data availability
All data generated or analyzed during this study are included in the manuscript and supporting files. Source data files have been provided for Figures 1, 3, 4, 5, 6, 8, 9 and 10.

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
