## [Decision Letter]

[Editors’ note: the authors submitted for reconsideration following the decision after peer review. What follows is the decision letter after the first round of review.]

Thank you for submitting your work entitled "HOPS recognizes each SNARE, assembling ternary *trans*-complexes for sudden fusion upon engagement with the 4th SNARE" for consideration by *eLife*. Your article has been reviewed by three peer reviewers, one of whom is a member of our Board of Reviewing Editors, and the evaluation has been overseen by a Senior Editor. The following individuals involved in review of your submission have agreed to reveal their identity: Josep Rizo (Reviewer #3).

Our decision has been reached after consultation between the reviewers. Based on these discussions and the individual reviews below, we regret to inform you that your work will not be considered further for publication in *eLife*.

As you will find from the attached review report, all three reviewers like your methodology and elegant data. However, reviewers are not convinced of the major conclusion that vacuolar SNAREs assemble via multiple parallel pathways. It remains possible that the SNARE assembly under your experimental conditions occurs via the template complex intermediate.

Reviewer #1:

SNARE proteins mediate membrane fusion through their coupled folding and assembly into a four-helix bundle. However, the pathway of SNARE assembly is unclear. In this manuscript, Song et al. addressed this issue using vacuolar fusion machinery and the reconstituted liposome fusion assay. They reconstituted Q-SNAREs and R-SNAREs into different liposomes and detected SNARE-mediated content mixing in the presence of HOPS and Rab proteins using a fluorescence assay previously developed in the same lab. To dissect the SNARE assembly pathway, they varied the species of Q-SNAREs (Qa, Qb, or Qc) anchored on the Q-liposomes and added HOPS and other soluble Q-SNAREs in the solution in different orders. They found that HOPS is generally required for liposome fusion, expect for Qabc-liposomes. Interestingly, they discovered a burst of liposome fusion when Q-liposomes containing any combinations of two Q-SNARE proteins preincubated with HOPS was mixed with the corresponding fourth SNARE protein in the solution. In addition, they observed that HOPS also binds Qb SNARE, suggesting that HOPS bind all four vacuolar SNAREs. Based on these findings, the authors proposed that the HOPS-chaperoned SNARE assembly follows several parallel pathways in terms of the order of SNARE addition into the SNARE complex. Accordingly, HOPS helps form a series of activated intermediates containing any combinations of two Q SNAREs and the R SNARE that facilitates incorporation of the third Q-SNARE. Overall, the work fills in an important gap in our understanding on SNARE assembly and experimental results are beautiful. However, although the conclusion on the parallel SNARE assembly is possible, alternative explanations exist, as detailed below. Therefore, the manuscript should be revised to consider these alternative pathways of SNARE assembly and new experiments are likely required to clarify the major conclusion.

1) The current assay does not directly detect SNARE assembly. As a result, SNARE assembly can only be inferred from the rate of membrane fusion. Earlier work from groups of Wickner and Hughson suggests that Vps33 simultaneously binds Qa- and R-SNAREs to serve as an essential intermediate (the template complex) for SNARE assembly. It appears that this model can still explain the experimental results shown in this work. For example, during pre-incubation, HOPS complexes start to tether Q- and R-liposomes and bind R-SNAREs in the Vps33 subunit, which serves as a rate-limiting step for SNARE assembly. Then Qa-SNARE quickly joins the Vps33-R complex to form the template complex. Finally, other Q-SNAREs rapidly bind the templated SNAREs to complete SNARE assembly and membrane fusion. In this pathway, the template complex becomes an obligate intermediate for SNARE assembly.

2) It is clear that HOPS binds each of the four vacuolar SNARE proteins. However, it is unclear that HOPS can simultaneously bind any two Q-SNAREs. A pull-down assay may suffice to clarify the different binding modes.

3) Do Qa-SNAREs form any binary complex? In all diagrams of Q-liposomes containing two Q-SNAREs, the two Q-SNAREs appear to form a dimer (e.g., Figure 1C). Is Qabc the only tripartite SNARE complex?

4) The authors frequently mentioned "*trans*-complex" and "*trans*-SNARE complex" in the text. It appears that all primed vesicles are mediated by some sort of "*trans*-complexes". To avoid confusion, shall "*trans*-SNARE complex" be used throughout the text?

Reviewer #2:

The authors use reconstituted proteoliposomes to continue their longstanding investigation of HOPS/SNARE-mediated vacuolar fusion. The most surprising and potentially interesting result is that R-SNARE liposomes, when incubated with HOPS and Q-SNARE liposomes containing any two Q-SNAREs, fuse extremely rapidly when the third Q-SNARE is added in soluble form. This appears to suggest that HOPS can organize any three SNAREs (provided one of them is the R-SNARE) into a membrane-bridging complex that allows rapid assimilation of the fourth SNARE and thereby the completion of zippering and membrane fusion.

My main concern, as relates to suitability for *eLife*, is whether there is sufficient mechanistic insight. The central observation is fascinating but I have trouble picturing how, on the molecular level, HOPS could actually accomplish the feat of 'mediating the assembly of a versatile set of activated fusion intermediates'. More insight into the nature of these intermediates, if it could be provided, would be an exciting addition.

Reviewer #3:

This paper describes an interesting study of how the HOPS tethering complex coordinates assembly of the yeast vacuolar SNARE complex. Previous work had shown that HOPS strongly stimulated fusion between liposomes containing the R-SNARE and Ypt7 with liposomes containing the three Q SNAREs and Ypt7, in part through the templating function of the Vps33 subunit of HOPS, which binds to the Qa and R SNAREs. However, HOPS could largely be replaced in these fusion assays by an artificial tether consisting of GST fused to a PX domain that binds to PI3P incorporated in both liposome populations. These results raised a key question: to what extent HOPS functions primarily to tether the two membranes, while HOPS-SNARE interactions play only a secondary, non-essential role? In this paper, the authors used three different types of liposomes containing pairs of Q SNAREs, adding the third Q SNARE in soluble form. They show that HOPS stimulated fusion of these liposomes with R-liposomes, but GST-PX was able to support only very slow fusion involving QaQb-liposomes, and no fusion for the other two types of double Q-SNARE liposomes. The paper further shows that HOPS interacts with each of the individual SNAREs and that pre-incubating the double Q-SNARE liposomes with HOPS leads to fast fusion with R-liposomes upon addition of the soluble Q SNARE. These results lead to a model whereby HOPS contains binding sites for the four SNAREs and can help to assemble distinct types of intermediates that contain different combinations of three SNAREs and can readily assemble with the fourth SNARE to form active *trans*-SNARE complexes. While the physiological relevance of this 'multi-templating' function of HOPS remains to be demonstrated, it makes a lot of sense, and the results presented in this paper constitute a framework to pursue this demonstration once the underlying HOPS-SNARE interactions are better characterized. I believe that these results will be of strong interest to a wide audience and have a few suggestions for revisions.

1) The authors showed earlier that 3Q-SNARE liposomes do not really need HOPS to fuse with R liposomes. Are the results obtained in this paper more relevant? Because vacuolar membranes contain all 4 SNAREs and these SNAREs can likely form *cis*-four-helix bundles with different compositions, Sec17 and Sec18 are critical to disassemble these *cis* complexes. After disassembly, HOPS likely plays a key role in 'catching' the individual SNAREs and placing them in correct orientations before they can re-assemble into *cis* complexes. Thus, it would be very informative if the authors analyze the effects of Sec17 and Sec18 in the assays presented in this paper. Although I strongly encourage the authors to perform these experiments, if they feel that a detailed analysis would be outside the scope of this paper, they should at least discuss in more the mechanistic implications of their findings as outline above.

[Editors’ note: further revisions were suggested prior to acceptance, as described below.]

Thank you for submitting your article "HOPS recognizes each SNARE, assembling ternary *trans*-complexes for rapid fusion upon engagement with the 4th SNARE" for consideration by *eLife*. Your article has been reviewed by three peer reviewers, one of whom is a member of our Board of Reviewing Editors, and the evaluation has been overseen by Vivek Malhotra as the Senior Editor. The following individuals involved in review of your submission have agreed to reveal their identity: Mary Munson (Reviewer #4).

The reviewers have discussed the reviews with one another and the Reviewing Editor has drafted this decision to help you prepare a revised submission.

Summary:

The authors make the intriguing discovery that appropriate pre-incubation of SNARE/Rab bearing liposomes and HOPS can yield 'intermediates' that fuse extremely rapidly when the remaining SNARE(s) are added in soluble form. This points to the ability of HOPS to organize the apposed membranes and SNARE proteins, so that fusion can occur efficiently. They found that HOPS also binds the Qb SNARE, an interaction that was previously undiscovered. Combined with previous findings, the result suggests that HOPS can bind each of the SNAREs, which may promote SNARE assembly and membrane fusion. Overall, the experiments are well designed and performed, which yield data of high quality.

The reviewers are curious about the molecular nature of the 'intermediates' and the molecular mechanism underlying the rapid fusion. The data suggest that simultaneous binding of multiple SNAREs to the HOPS complex help initiate SNARE assembly, likely by the SM subunit Vps33. Similar mechanisms of SNARE recruitment and chaperoned assembly have been observed in many SNARE-mediated fusion systems. However, this mechanism is not clearly spelled out in the manuscript, which might put burden on readers to rationalize the abundant experimental observations. The reviewers understand that the data here may not support a unique mode at this stage. But a simplistic model that maximally explains current data with minimal assumptions will greatly help.

Essential revisions:

1) It has long been recognized that individual SNAREs need to be recruited to the fusion site to initiate SNARE assembly and subsequent membrane fusion. The observation that HOPS may simultaneously bind multiple SNAREs is consistent with this view. The authors suggest that there are multiple "activated" *trans* complexes. However, it is unclear whether these activated complexes simply help recruit SNAREs or play a conceptually new role in SNARE assembly. The very similar kinetics displayed by several different constellations of SNAREs is an especially intriguing feature. What could it mean? Does it signify a common pathway, perhaps involving Vps33 templating as a common rate-limiting step? Schematic diagrams in a new figure are recommended to illustrate the three activated complexes and potential common SNARE assembly pathway.

2) The idea that HOPS recruits all SNAREs together is compelling, and it would be helpful to a broader readership to speculate and generalize these findings to other multi-subunit tethering complexes and their partner SNAREs.

3) A major finding reported here is that HOPS binds Qb-SNAREs. It is however somewhat surprising, given the intensity with which HOPS and its cognate SNAREs have been studied, that this discovery wasn't made earlier. Perhaps this has to do with the decision to use liposome floatation as a binding assay. Given that the liposomes lack the lipids that HOPS and SNAREs are known to bind, what do the authors imagine is the role they are playing? If each SNARE binds HOPS, why do the four SNAREs together bind less well? Given that binding to Qc, at least, involves a domain other than the SNARE motif, shouldn't binding to 4-SNARE liposomes be at least that good? Can anything meaningful be said about the actual affinity for this newly-reported interaction?

4) The reviewers were somewhat perplexed by the "saturability" experiments (final paragraph of the Introduction section, subsection “HOPS function is saturable for each Q-SNARE”, Figure 10), which are presented as a complement to the HOPS binding assays. The authors argue for a qualitative difference between HOPS and PEG, but it seems to me that it is also plausible that they are observing a quantitative difference. That is, because HOPS is more efficient than PEG, the assay is saturated at all tested values. Higher concentrations would surely reveal that PEG can saturate the assay too, whereas lower concentrations would reveal a range in which HOPS too would fail to saturate. Given these considerations, the authors should address the concern that their saturability experiments do not, in fact, represent independent evidence for specific SNARE binding sites.

5) The authors write: "In an earlier model sub-reaction (Orr et al., 2017), proteoliposomes bearing Ypt7 and R-SNARE were incubated with the 3 soluble Q-SNAREs and HOPS, then re-isolated by floatation. HOPS was required for the association of each of the Q-SNAREs, and each Q-SNARE depended on the other two for its HOPS-dependent membrane association." If HOPS binds independently to each SNARE, why does each Q-SNARE depend on the other two for HOPS-dependent membrane association?

6) The authors state that "one limitation (of R-Qa-SM as the 'unique and committed step') is that 4-SNARE assembly of R- with Q-SNAREs can proceed without SM function as long as there is tethering and the three Q-SNAREs are pre-assembled (Figure 1A, B)." Is this physiologically relevant? That is, how, on a vacuole, would the three Q-SNAREs pre-assemble without forming *cis* complexes with the R-SNARE?

7) How do the authors exclude a model in which HOPS:R-SNARE association is slow and rate-limiting for the formation of the rapid-fusion intermediate(s)? Could this explain why different Q-SNARE combinations behave almost indistinguishably?

---

## [Author Response]

[Editors’ note: the authors resubmitted a revised version of the paper for consideration. What follows is the authors’ response to the first round of review.]

Reviewer #1:SNARE proteins mediate membrane fusion through their coupled folding and assembly into a four-helix bundle. However, the pathway of SNARE assembly is unclear. In this manuscript, Song et al. addressed this issue using vacuolar fusion machinery and the reconstituted liposome fusion assay. They reconstituted Q-SNAREs and R-SNAREs into different liposomes and detected SNARE-mediated content mixing in the presence of HOPS and Rab proteins using a fluorescence assay previously developed in the same lab. To dissect the SNARE assembly pathway, they varied the species of Q-SNAREs (Qa, Qb, or Qc) anchored on the Q-liposomes and added HOPS and other soluble Q-SNAREs in the solution in different orders. They found that HOPS is generally required for liposome fusion, expect for Qabc-liposomes. Interestingly, they discovered a burst of liposome fusion when Q-liposomes containing any combinations of two Q-SNARE proteins preincubated with HOPS was mixed with the corresponding fourth SNARE protein in the solution. In addition, they observed that HOPS also binds Qb SNARE, suggesting that HOPS bind all four vacuolar SNAREs. Based on these findings, the authors proposed that the HOPS-chaperoned SNARE assembly follows several parallel pathways in terms of the order of SNARE addition into the SNARE complex. Accordingly, HOPS helps form a series of activated intermediates containing any combinations of two Q SNAREs and the R SNARE that facilitates incorporation of the third Q-SNARE. Overall, the work fills in an important gap in our understanding on SNARE assembly and experimental results are beautiful. However, although the conclusion on the parallel SNARE assembly is possible, alternative explanations exist, as detailed below. Therefore, the manuscript should be revised to consider these alternative pathways of SNARE assembly and new experiments are likely required to clarify the major conclusion.

We've added substantial data which both confirm that R‐ and Qa‐ can interact with HOPS to form a sudden‐fusion intermediate, and characterized this in several regards, but also directly show that the sudden fusion intermediate formed by HOPS with Ypt7/R and Ypt7/QbQc proteoliposomes in the complete absence of Qa entails HOPS‐dependent R- and Qb‐SNARE assembly into in the same complex (Figure 8). We've also added data showing that RQbQc complexes don't form spontaneously in detergent solution (Figure 7), underscoring that HOPS is assembling them. We've thus worked faithfully to "consider alternative pathways" and added substantial "new experiments" while directly addressing and further bolstering the R:Qa pathway.

1) The current assay does not directly detect SNARE assembly. As a result, SNARE assembly can only be inferred from the rate of membrane fusion.

We directly show by co‐immunoprecipitation that the rapid fusion intermediate which HOPS forms with Ypt7/R and Ypt7/QaQb proteoliposomes has Qa and R in the same complex (Figure 3), as expected; what's surprising and novel is that HOPS can also form a rapid‐fusion intermediate with Ypt7/R and Ypt7/QbQc in which Qb and R are shown to be in the same complex in the absence of Qa (Figure 8), even though SNAREs alone do not form RQbQc complex, as we also show (Figure 7). We don't know whether or not the 3 SNAREs are in a coiled‐coils configuration, perhaps stabilized by HOPS, or are directly bound to HOPS and not to each other.

Earlier work from groups of Wickner and Hughson suggests that Vps33 simultaneously binds Qa- and R-SNAREs to serve as an essential intermediate (the template complex) for SNARE assembly. It appears that this model can still explain the experimental results shown in this work. For example, during pre-incubation, HOPS complexes start to tether Q- and R-liposomes and bind R-SNAREs in the Vps33 subunit, which serves as a rate-limiting step for SNARE assembly. Then Qa-SNARE quickly joins the Vps33-R complex to form the template complex. Finally, other Q-SNAREs rapidly bind the templated SNAREs to complete SNARE assembly and membrane fusion. In this pathway, the template complex becomes an obligate intermediate for SNARE assembly.

Inherent to this formulation is the idea that R and Qa must both bind to Vps33 before Qb and Qc join with them to give a 4‐SNARE complex. This is fine for HOPS:R:QaQb and HOPS:R:QaQc. However, the HOPS + YR + YQbc complex does not include Qa, and we now show directly that there is a complex with Qb and R associated, either directly (SNARE to SNARE) or via their mutual affinities for HOPS. We emphasize throughout that we don't yet know whether the rapid‐fusion intermediate represents HOPS catalyzing the formation of ternary coiled‐coils, whether HOPS remains bound to stabilize otherwise unstable ternary SNARE intermediates, or whether the SNAREs are only bound to HOPS and not yet to each other, being triggered by the 4^th^ SNARE to be released into a coiled‐coil.

2) It is clear that HOPS binds each of the four vacuolar SNARE proteins. However, it is unclear that HOPS can simultaneously bind any two Q-SNAREs. A pull-down assay may suffice to clarify the different binding modes.

This is an excellent suggestion, yet a lot of focused effort in our lab has failed to prove that the same HOPS molecules bound to one SNARE are also bound to another (or, that they're not). HOPS has direct affinity for membrane lipids too (e.g. Orr et al., 2014, Figure 4), and we suspect that the tight binding (e.g. nM) of HOPS to a single liposome‐bound SNARE is the product of its (e.g. μM) affinity for the SNARE and (e.g. mM) affinity for lipid. When we add a second, soluble SNARE, HOPS doesn't cause this second soluble SNARE to bind stably enough to be assayed by flotation. Our efforts to achieve this, or conversely to show displacement of binding to one SNARE by another, are ongoing but not yet successful.

3) Do Qa-SNAREs form any binary complex? In all diagrams of Q-liposomes containing two Q-SNAREs, the two Q-SNAREs appear to form a dimer (e.g., Figure 1C). Is Qabc the only tripartite SNARE complex?

We have no data on whether 2Q‐SNARE proteoliposomes have these SNAREs bound to the membrane but not to each other, or in QaQb, QaQc, or QbQc complexes, but you're right about our diagrams being misleading. We've now redrawn 'em all to not show such associations, though we've left the 3Q complexes cartooned in Figure 1A, B, as 3Q complexes clearly are stable, as shown before (Fukuda et al., 2000) and now again in Figure 7A, lane 6. In detergent, there are tripartite complexes seen as long as Qa is present; we now show this directly in Figure 7.

4) The authors frequently mentioned "trans-complex" and "trans-SNARE complex" in the text. It appears that all primed vesicles are mediated by some sort of "trans-complexes". To avoid confusion, shall "trans-SNARE complex" be used throughout the text?

A *trans*‐complex of a sudden fusion intermediate might have the SNAREs in a tripartite (or bipartite?) coiled‐coils structure with each other, stabilized by being bound up somehow with HOPS, or the SNAREs (anchored to separate bilayers) could only be part of the complex because they're bound to their sites on HOPS. Either would be a *trans* complex. We've now addressed this explicitly, and reserve the term "*trans*‐SNARE complex" for when the SNAREs are reasonably inferred to be directly bound up with each other. In subsection “HOPS assembles R- and Qa-SNARE fusion intermediates”, where we first characterize (Figure 3) a rapid‐fusion complex, we write "We refer to these as "*trans*-complexes", since they include two proteins anchored to different membranes, and reserve the term "*trans*-SNARE complex" for when the anchored SNAREs themselves are likely in a coiled coils complex with each other."

This is the first report of a protein (6‐subunit HOPS) that recognizes all 4 SNAREs, and provides a lot of information on how HOPS catalyzes their assembly. We have put a huge effort into this study, and hope that its novelty and depth with permit its acceptance for *eLife*.

Reviewer #2:The authors use reconstituted proteoliposomes to continue their longstanding investigation of HOPS/SNARE-mediated vacuolar fusion. The most surprising and potentially interesting result is that R-SNARE liposomes, when incubated with HOPS and Q-SNARE liposomes containing any two Q-SNAREs, fuse extremely rapidly when the third Q-SNARE is added in soluble form. This appears to suggest that HOPS can organize any three SNAREs (provided one of them is the R-SNARE) into a membrane-bridging complex that allows rapid assimilation of the fourth SNARE and thereby the completion of zippering and membrane fusion.My main concern, as relates to suitability for eLife, is whether there is sufficient mechanistic insight. The central observation is fascinating but I have trouble picturing how, on the molecular level, HOPS could actually accomplish the feat of 'mediating the assembly of a versatile set of activated fusion intermediates'. More insight into the nature of these intermediates, if it could be provided, would be an exciting addition.

Thank you – we've now provided substantial additional data which tells us more about the mechanism. Most notably, though SNAREs alone will not form stable complexes in detergent without Qa (new Figure 7), we've now added Figure 8 which shows that HOPS mediates formation of a stable sudden‐fusion complex which includes R‐ and Qb‐SNAREs anchored in *trans* in the complete absence of Qa, even though SNAREs alone don't form complexes without Qa. This very directly expands our view of catalyzed SNARE assembly, physically and functionally, showing that the affinity of HOPS for Qb is functionally relevant. We also show that HOPS will assemble separately proteoliposome-anchored R‐ and Qa‐SNAREs alone into a sudden‐fusion complex, fusing upon encountering Qb and Qc, and characterize this sudden‐fusion intermediate, but have not yet been able to purify the complex and study it in isolation. This is a central (future) goal of my lab.

This is the first report of a protein (6‐subunit HOPS) that recognizes all 4 SNAREs, and provides a lot of information on how HOPS catalyzes their assembly. We show in Figure 1 that the SM function of HOPS isn't needed when the three Q‐SNAREs are assembled; rather, it's specific function beyond simply tethering is catalyzing the entry of each SNARE into the system. There's nothing like this for other systems, no sign of specific Qb or Qc recognition in the neuronal system, for example (where they're together as one protein, SNAP‐25). We feel confident in saying that this is important for the membrane fusion field, and hope that you find this too.

Reviewer #3:This paper describes an interesting study of how the HOPS tethering complex coordinates assembly of the yeast vacuolar SNARE complex. Previous work had shown that HOPS strongly stimulated fusion between liposomes containing the R-SNARE and Ypt7 with liposomes containing the three Q SNAREs and Ypt7, in part through the templating function of the Vps33 subunit of HOPS, which binds to the Qa and R SNAREs. However, HOPS could largely be replaced in these fusion assays by an artificial tether consisting of GST fused to a PX domain that binds to PI3P incorporated in both liposome populations. These results raised a key question: to what extent HOPS functions primarily to tether the two membranes, while HOPS-SNARE interactions play only a secondary, non-essential role? In this paper, the authors used three different types of liposomes containing pairs of Q SNAREs, adding the third Q SNARE in soluble form. They show that HOPS stimulated fusion of these liposomes with R-liposomes, but GST-PX was able to support only very slow fusion involving QaQb-liposomes, and no fusion for the other two types of double Q-SNARE liposomes. The paper further shows that HOPS interacts with each of the individual SNAREs and that pre-incubating the double Q-SNARE liposomes with HOPS leads to fast fusion with R-liposomes upon addition of the soluble Q SNARE. These results lead to a model whereby HOPS contains binding sites for the four SNAREs and can help to assemble distinct types of intermediates that contain different combinations of three SNAREs and can readily assemble with the fourth SNARE to form active trans-SNARE complexes. While the physiological relevance of this 'multi-templating' function of HOPS remains to be demonstrated, it makes a lot of sense, and the results presented in this paper constitute a framework to pursue this demonstration once the underlying HOPS-SNARE interactions are better characterized. I believe that these results will be of strong interest to a wide audience and have a few suggestions for revisions.

We've strengthened our findings with the HOPS:R:Qa rapid‐fusion intermediate, and added data on the HOPS:R:QbQc rapid‐fusion intermediate which demonstrate that Qb and R are in the same complex.

1) The authors showed earlier that 3Q-SNARE liposomes do not really need HOPS to fuse with R liposomes. Are the results obtained in this paper more relevant?

There is little data evaluating whether the 3 Q‐SNAREs remain together through multiple fusion cycles or dissociate, spontaneously or by Sec17/Sec18/ATP, but we believe that the discovery of rapid‐fusion intermediates which can contain novel SNARE combinations (R, Qb, Qc) which will not stably associate in detergent (new Figure 7B, C) is of great mechanistic interest.

Because vacuolar membranes contain all 4 SNAREs and these SNAREs can likely form cis-four-helix bundles with different compositions, Sec17 and Sec18 are critical to disassemble these cis complexes. After disassembly, HOPS likely plays a key role in 'catching' the individual SNAREs and placing them in correct orientations before they can re-assemble into cis complexes. Thus, it would be very informative if the authors analyze the effects of Sec17 and Sec18 in the assays presented in this paper.

There's much, much more to do, but we've now added Figure 4 which shows that Sec17/18/ATP do not disassemble or inhibit either the HOPS:R:QaAb or HOPS:R:QbQc rapid‐fusion intermediates.

[Editors’ note: what follows is the authors’ response to the second round of review.]

The reviewers are curious about the molecular nature of the 'intermediates' and the molecular mechanism underlying the rapid fusion. The data suggest that simultaneous binding of multiple SNAREs to the HOPS complex help initiate SNARE assembly, likely by the SM subunit Vps33. Similar mechanisms of SNARE recruitment and chaperoned assembly have been observed in many SNARE-mediated fusion systems. However, this mechanism is not clearly spelled out in the manuscript, which might put burden on readers to rationalize the abundant experimental observations. The reviewers understand that the data here may not support a unique mode at this stage. But a simplistic model that maximally explains current data with minimal assumptions will greatly help.

Thank you, we now provide such a model. It includes the fact that the R- and Qa-SNARE domains, as bound to Vps33 in the crystal structures of Baker et al., 2015, have their largely apolar surfaces facing into their grooves on Vps33, the very same surfaces which face inward on the 4-SNARE bundle. We write:

"We suggest a working model (Figure 11). The binding sites on HOPS for each of the 4 individual SNAREs mediate the initial HOPS:SNARE associations (Step A). […] Each intermediate, whether HOPS:R:Qa (alone or with Qb or Qc) or HOPS:R:Qb:Qc, and whether the SNAREs remain bound to their initial sites on HOPS or have begun coiled-coils assembly, is poised to accept the missing SNAREs (Step C) for very rapid fusion."

Essential revisions:1) It has long been recognized that individual SNAREs need to be recruited to the fusion site to initiate SNARE assembly and subsequent membrane fusion. The observation that HOPS may simultaneously bind multiple SNAREs is consistent with this view. The authors suggest that there are multiple "activated" trans complexes. However, it is unclear whether these activated complexes simply help recruit SNAREs or play a conceptually new role in SNARE assembly. The very similar kinetics displayed by several different constellations of SNAREs is an especially intriguing feature. What could it mean? Does it signify a common pathway, perhaps involving Vps33 templating as a common rate-limiting step? Schematic diagrams in a new figure are recommended to illustrate the three activated complexes and potential common SNARE assembly pathway.

We have now included a working model, as above; unless the binding site for Qb is also on Vps33 (possible, but without experimental foundation), templating for the R:QbQc ternary intermediate would have to extend to other HOPS subunits as well. We're trying to not present a model which goes too far beyond the data, and so our model is just schematic.

2) The idea that HOPS recruits all SNAREs together is compelling, and it would be helpful to a broader readership to speculate and generalize these findings to other multi-subunit tethering complexes and their partner SNAREs.

We've now added brief discussion of whether and where our findings of physical and functional recognition of all 4 of the SNAREs may apply in other fusion reactions; we're not aware of this being clearly resolved in other fusion systems.

3) A major finding reported here is that HOPS binds Qb-SNAREs. It is however somewhat surprising, given the intensity with which HOPS and its cognate SNAREs have been studied, that this discovery wasn't made earlier. Perhaps this has to do with the decision to use liposome floatation as a binding assay. Given that the liposomes lack the lipids that HOPS and SNAREs are known to bind, what do the authors imagine is the role they are playing?

For reasons you noted, we don't expect that it's recognition of the PC headgroup. Rather, we speculate that there is some very low (e.g. mM) binding affinity for the headgroup/acyl chain interface; if HOPS affinity for Qb was just e.g. μM, and HOPS affinity for PC is just e.g. mM, then HOPS affinity for Qb bound to PC may be μM x millimolar = nanomolar. We've added this concept to the text.

If each SNARE binds HOPS, why do the four SNAREs together bind less well?

We now emphasize in the text that the apolar surfaces of R and Qa, which alone will face into their grooves on Vps33, are hidden by facing into the 4-SNARE complex. The 4-SNARE complex may therefore may only bind HOPS by the HOPS affinities for Qb or Qc.

Given that binding to Qc, at least, involves a domain other than the SNARE motif, shouldn't binding to 4-SNARE liposomes be at least that good?

Yes, or the Qc N-domain, where HOPS binds (Stroupe et al., 2006), may be somewhat occluded for HOPS binding when in complex with the other SNAREs, which themselves possess N-domains too. We comment on this in the text now.

Can anything meaningful be said about the actual affinity for this newly-reported interaction?

We don't yet have binding constants for these interactions.

4) The reviewers were somewhat perplexed by the "saturability" experiments (final paragraph of the Introduction section, subsection “HOPS function is saturable for each Q-SNARE”, Figure 10), which are presented as a complement to the HOPS binding assays. The authors argue for a qualitative difference between HOPS and PEG, but it seems to me that it is also plausible that they are observing a quantitative difference. That is, because HOPS is more efficient than PEG, the assay is saturated at all tested values. Higher concentrations would surely reveal that PEG can saturate the assay too, whereas lower concentrations would reveal a range in which HOPS too would fail to saturate. Given these considerations, the authors should address the concern that their saturability experiments do not, in fact, represent independent evidence for specific SNARE binding sites.

Thanks for this very thoughtful critique. We've now explained this more clearly in the text:

"In short, only tethered proteoliposomes will assemble *trans*-SNARE complexes and proceed to fuse. Once HOPS or PEG has tethered the membranes, *trans*-SNARE assembly can begin. If HOPS had no function beyond tethering, then membranes tethered by HOPS or PEG would have the same Km for each SNARE. However, when HOPS, which can recognize each SNARE, performs the tethering, we find that fusion has a far lower Km for each SNARE than when tethering is through PEG, which cannot recognize SNAREs. This indicates that HOPS not only functions by tethering but also by its recognition of each individual SNARE."

5) The authors write: "In an earlier model sub-reaction (Orr et al., 2017), proteoliposomes bearing Ypt7 and R-SNARE were incubated with the 3 soluble Q-SNAREs and HOPS, then re-isolated by floatation. HOPS was required for the association of each of the Q-SNAREs, and each Q-SNARE depended on the other two for its HOPS-dependent membrane association." If HOPS binds independently to each SNARE, why does each Q-SNARE depend on the other two for HOPS-dependent membrane association?

In the study of Orr et al., we were measuring SNARE complex assembly on liposomes bearing only one SNARE, and thus this was in the absence of fusion, i.e., those were *cis*-SNARE complexes. We've now added:

"HOPS supported the assembly of those *cis*-complexes, dependent on all 4 SNAREs; *cis*-complexes with 3 SNAREs are presumably not stable. In our current study, we examine intermediates in *trans*-SNARE assembly which lack one or the other of the 3 Q-SNAREs, and report the existence of rapid-fusion intermediates for each of the 3 "missing" Q-SNAREs, including Qa."

6) The authors state that "one limitation (of R-Qa-SM as the 'unique and committed step') is that 4-SNARE assembly of R- with Q-SNAREs can proceed without SM function as long as there is tethering and the three Q-SNAREs are pre-assembled (Figure 1A, B)." Is this physiologically relevant? That is, how, on a vacuole, would the three Q-SNAREs pre-assemble without forming cis complexes with the R-SNARE?

We'd written this poorly, not making clear the point of the sentence. We now replace this sentence with:

“One limitation to the concept that R and Qa can only associate during templating by an SM protein is that 4-SNARE assembly of R- with Q-SNAREs can proceed without SM function as long as there is tethering and the three Q-SNAREs are pre-assembled (Figure 1A and B). In this context, the Qb and Qc SNAREs which are associated with Qa may substitute for the SM templating function. It is unclear whether the three Q-SNAREs ever physiologically pre-assemble in the presence of Sec17, Sec18, and ATP.”

7) How do the authors exclude a model in which HOPS:R-SNARE association is slow and rate-limiting for the formation of the rapid-fusion intermediate(s)? Could this explain why different Q-SNARE combinations behave almost indistinguishably?

We don't address which step is rate-limiting in forming each of the 3 rapid-fusion intermediates, and HOPS:R-SNARE association could be rate limiting in their formation. Our point is that each of these intermediates exist, including the most unexpected one lacking Qa, and that each of these ternary intermediates gives very rapid fusion when the missing sQ-SNARE is added.